# Surface hydrophobization of hydrogels via interface dynamics-induced network reconfiguration

Bo Yi[1,2,8], Tianjie Li [3,8], Boguang Yang[4,8], Sirong Chen[1,2], Jianyang Zhao[1,2], Pengchao Zhao[1,2,5,6], Kunyu Zhang [1,2,5,6], Yi Wang [3 ✉], Zuankai Wang [7 ✉] & Liming Bian [1,2,5,6 ✉]

Effective and easy regulation of hydrogel surface properties without changing the overall chemical composition is important for their diverse applications but remains challenging to achieve. We report a generalizable strategy to reconfigure hydrogel surface networks based on hydrogel–substrate interface dynamics for manipulation of hydrogel surface wettability and bioadhesion. We show that the grafting of hydrophobic yet flexible polymeric chains on mold substrates can significantly elevate the content of hydrophobic polymer backbones and reduce the presence of polar groups in hydrogel surface networks, thereby transforming the otherwise hydrophilic hydrogel surface into a hydrophobic surface. Experimental results show that the grafted highly dynamic hydrophobic chains achieved with optimal grafting density, chain length, and chain structure are critical for such substantial hydrogel surface network reconfiguration. Molecular dynamics simulations further reveal the atomistic details of the hydrogel network reconfiguration induced by the dynamic interface interactions. The hydrogels prepared using our strategy show substantially enhanced bioadhesion and transdermal delivery compared with the hydrogels of the same chemical composition but fabricated via the conventional method. Our findings provide important insights into the dynamic hydrogel–substrate interactions and are instrumental to the preparation of hydrogels with custom surface properties.

Hydrogels, polymer networks swollen by an extremely high content of water, have found broad applications ranging from cell culture, drug delivery, and tissue repair to wearable electronics and water harvesting[1–5]. While the bulk properties of hydrogels have received extensive research attention[6–8], the hydrogel surface properties are also crucial for their efficient function and specific applications. For example, the adhesion of hydrogels depends on the interfacial interaction between the hydrogel surface and substrates, and the rational design of hydrogel surface functional groups and network structure can strengthen the interfacial interaction, thereby influencing the

[1]School of Biomedical Sciences and Engineering, Guangzhou International Campus, South China University of Technology, Guangzhou 511442, PR China. [2]National Engineering Research Center for Tissue Restoration and Reconstruction, South China University of Technology, Guangzhou 510006, PR China. [3]Department of Physics, The Chinese University of Hong Kong, Hong Kong 999077, PR China. [4]Department of Orthopaedic and Traumatology, The Chinese University of Hong Kong, Hong Kong 999077, PR China. [5]Guangdong Provincial Key Laboratory of Biomedical Engineering, South China University of Technology, Guangzhou 510006, PR China. [6]Key Laboratory of Biomedical Materials and Engineering of the Ministry of Education, South China University of Technology, Guangzhou 510006, PR China. [7]Department of Mechanical Engineering, The Hong Kong Polytechnic University, Hong Kong 999077, PR China. [8]These authors contributed equally: Bo Yi, Tianjie Li, Boguang Yang. ✉e-mail: yiwang@cuhk.edu.hk; zk.wang@polyu.edu.hk; bianlm@scut.edu.cn

adhesion force[9,10]. In particular, the wettability of the hydrogel surface is of great importance to the adhesion of hydrogels on wet biological tissues because the physiological fluids at the interface substantially hinder the close contact of hydrogels with tissues[11,12]. A hydrophobic hydrogel surface can displace interfacial water and improve the wet adhesion of hydrogels[13]. Chemical strategies based on chemical modifications of the monomers/polymers[14,15] and nonchemical strategies based on methods including surface coating techniques[16,17], solvent exchange methods[13,18], and physical stimuli conditioning[19,20] have been extensively explored for the surface property management of hydrogels. However, these strategies typically change the chemical composition and physical structures of hydrogels either in their entirety or at the interface and may have undesirable effects on their bulk properties, such as deteriorated biocompatibility due to the employed chemical reagents or solvents[21], mismatched mechanical properties at hydrogel interfaces due to dense and hard surface layers[22], and compromised transparency due to colored ingredients or opaque surface coatings[23]. Synthetic hydrogel networks generally consist of crosslinked polymer carbon backbones and appended polar groups to provide structural integrity and retain water molecules, respectively.

Herein, we propose a simple yet versatile strategy by inducing surface network reconfiguration to regulate hydrogel surface wettability and bioadhesion without altering their bulk chemical and physical properties. We hypothesize that the interface dynamics arising from hydrophobic polymeric chains grafted on the mold surfaces can induce the surface network reconfiguration of hydrogels during the gelation process by weak but dense noncovalent interactions such as hydrophobic interactions and van der Waals forces. Because the regulation of hydrogel surface properties is based on the hydrogel–substrate interface dynamics, we named this hydrogel surface regulation strategy the interface dynamics-induced network reconfiguration (DNR). The polymeric chain-based dynamic hydrogel–substrate interface was designed on the basis of two criteria: (i) the polymeric chains should be covalently anchored on the mold surfaces at one end while being highly flexible at the other to effectively interact with the hydrogel surface networks; and (ii) the polymeric chains should be largely chemically inert to the monomers/polymers of hydrogels to avoid the formation of strong chemical bonds such as covalent bonds and ionic bonds.

## Results and discussion

### Interface-mediated dynamic network reconfiguration (DNR) impacts hydrogel surface wettability

Under these design principles, we used the DNR strategy to manipulate the surface wettability of hydrogels from intrinsically hydrophilic to highly hydrophobic states without changing the chemical composition and bulk properties of hydrogels (Fig. 1a). We first covalently grafted flexible hydrophobic silicone chains[24,25] to the glass mold surface to generate a nanoscale layer of structurally dynamic polymeric coating (Supplementary Fig. 1). During the gelation process, the grafted silicone chains on the mold substrate could interact with the hydrophobic polymerizing carbon backbones of hydrogel surface networks to drive their conformational reconfiguration and reorientation at the substrate–hydrogel interface (i.e., DNR) (Fig. 1a). The mold-contacting surfaces of the obtained DNR hydrogels possessed a higher content of hydrophobic polymer backbones and showed significantly elevated surface hydrophobicity with a water contact angle (WCA) up to ~120° (Supplementary Fig. 2). In contrast, the hydrogels prepared with untreated molds (hereafter called conventional hydrogels) were naturally hydrophilic with a WCA of ~35°.

We first examined whether the mold surface hydrophobicity can directly contribute to the transformation of hydrogel surface wettability given that the silicone chain-grafted mold (named DNR mold) is hydrophobic (Supplementary Fig. 1). A series of untreated molds, including glass, poly(methyl methacrylate) (PMMA),

polytetrafluoroethylene (PTFE) and silicone rubber, with diverse wettability varying from hydrophilic to hydrophobic were used to prepare hydrogels. We tested the WCA of these hydrogels by employing the widely used poly(acrylic acid) (PAA) as the model hydrogel, and the mold-contacting surface of PAA hydrogels prepared using different molds were all hydrophilic with a WCA of 30–40° regardless of the mold substrate surface hydrophobicity (Supplementary Fig. 3a). For comparison, all the PAA hydrogels prepared with DNR molds had hydrophobic mold-contacting surfaces regardless of the mold substrate materials (Supplementary Fig. 3b). These results indicate the unlikely influence of mold hydrophobicity alone and the necessity of a dynamic hydrogel–substrate interface for the effective regulation of hydrogel surface wettability.

Previous work indicates that hydrogels prepared by a hydrophobic mold demonstrate a less crosslinked surface with lower modulus, compared to that prepared in a hydrophilic mold[26–28]. However, few prior studies have systemically investigated the impact of chain dynamics of mold surface-grafted hydrophobic polymers on the network reorganizations and wettability of hydrogel surfaces and associated mechanisms. Therefore, we believe that our work provide critical mechanistic insights to the hydrogel – mold surface interactions and associated changes in hydrogel surface properties from a molecular perspective, which is largely absent from prior literatures. Notably, the copolymerization of acrylic acid with a hydrophobic monomer, such as butyl acrylate, in a hydrophobic mold may also generate a hydrogel with a hydrophobic surface. However, the poor solubility and stability of hydrophobic monomers in aqueous solution would alter the hydrogel structure and result in an inhomogeneous and translucent hydrogel[29]. Moreover, the addition of hydrophobic monomers changes the chemical composition of the hydrogel and may have unfavorable effects on the hydrogel bulk properties[30]. Therefore, this situation is beyond our discussion and we focused on the DNR effect primarily of hydrogels with a single hydrophilic monomer.

To evaluate the general applicability of our DNR strategy, hydrogels based on another two polymers, poly(methacrylic acid) (PMAA) and polyacrylamide (PAAm), were prepared using the DNR mold and untreated mold (glass substrates were used unless otherwise specified), and all obtained DNR hydrogels showed significantly elevated surface hydrophobicity, as indicated by the substantial WCA difference between DNR and conventional hydrogels (Fig. 1b). The largest WCA difference (~110°) found in the PMAA hydrogels can be attributed to the additional hydrophobic methyl groups on the polymer backbone. Furthermore, the DNR hydrogels maintained surface hydrophobicity under a high humidity environment after 24 h, demonstrating the stability of surface hydrophobicity during hydrogel storage (Supplementary Fig. 4). Nevertheless, the WCA of the hydrogels slightly decreased with time during the contact angle measurement process, possibly because of the recovery of hydrophilic network components to the hydrogel surface induced by the persistent presence of probing water droplets (Supplementary Fig. 5 and Supplementary Movie 1)[20]. We further demonstrated the wide applicability of the DNR strategy by preparing double-network hydrogels, such as the PAA/gelatin and PAAm/alginate hydrogels, both of which demonstrated a significant DNR effect, as evidenced by the significant WCA difference (Supplementary Fig. 6). We also successfully fabricated DNR hydrogels with asymmetric surface hydrophobicity, also known as Janus hydrogels, by simply using silicone chain-grafted and untreated substrate surfaces on the opposing side of the mold during hydrogel preparation (Supplementary Fig. 7).

We next examined the importance of covalently grafting silicone chains to the mold surface by comparing it with a mold surface physically coated by silicone oil. The silicone oil used had the same composition of linear polydimethylsiloxane and the same high mobility as that of silicone chains. An ultrathin silicone oil layer was coated on the glass mold by a film applicator, and its WCA was similar to that of the

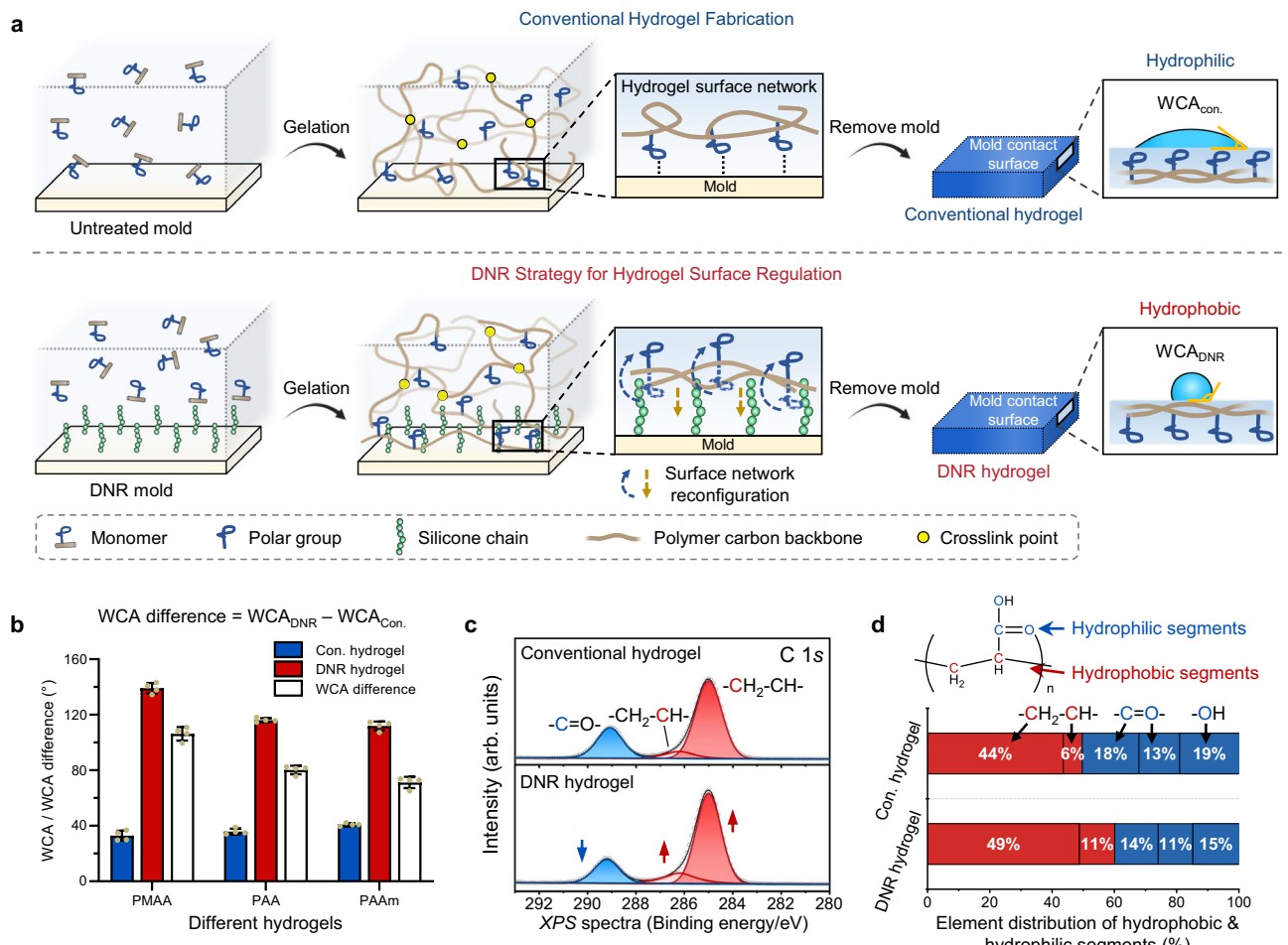

**Fig. 1 | Interface dynamics-induced network reconfiguration (DNR) regulates hydrogel surface wettability. a** Schematic illustration of the conventional hydrogel fabrication process and hydrogel surface wettability regulation by our DNR strategy. **b** Different hydrogels based on various polymers, including PMAA, PAA, and PAAm, were prepared using an untreated mold and a DNR mold (silicone chain-grafted mold), and the water contact angles (WCAs) of conventional (abbreviated as Con.) and DNR hydrogels were characterized. The WCA difference was calculated as follows: the WCA of the DNR hydrogel (WCA$_{DNR}$)−the WCA of the conventional hydrogel (WCA$_{Con.}$). PMAA poly(methacrylic acid), PAA poly(acrylic acid), PAAm polyacrylamide. **c** High-resolution *XPS* C 1 *s* spectra with deconvolution of polymer structures for conventional and DNR hydrogels. The term a.u. represents arbitrary units. **d** Structure of PAA with highlighted hydrophilic/hydrophobic segments and statistics of the element content distribution of polymer segments in conventional and DNR hydrogels. Values in (**b**) represent the mean ± SD; $n = 4$ independent samples.

DNR mold. The oil-coated mold and DNR mold were then used to prepare hydrogels (Supplementary Fig. 8). Although the hydrogels obtained from both molds showed hydrophobicity, the WCA value of the hydrogel prepared with the oil-coated mold was much lower and declined much faster compared with that of the DNR hydrogel (Supplementary Discussion 1 and Supplementary Movie 2). Moreover, the silicone oil would be entrained by the hydrogel upon extraction from the mold, resulting in a dramatic decrease in the WCA of the oil-coated mold. We also determined that the surface roughness of the DNR mold had a negligible influence on the hydrogel surface wettability (Supplementary Fig. 9).

## Surface chemistry characterization reveals the hydrogel surface network reconfiguration

To validate our hypothesis that polymeric chain-induced network reconfiguration is responsible for the increased hydrogel surface hydrophobicity, we analyzed the surface chemistry of conventional and DNR hydrogels using PAA as the model hydrogel. We first performed attenuated total reflection Fourier transform infrared spectroscopy (ATR-FTIR) on the two hydrogels (Supplementary Fig. 10a). The IR spectra confirmed the identical chemical composition of the two

hydrogels. However, compared with the conventional hydrogel, the DNR hydrogel presented stronger absorption peaks at 1173 cm$^{-1}$ and 806 cm$^{-1}$ assigned to C-C stretching and C-H deformation, respectively, and a weaker absorption band of O-H stretching at 3096 cm$^{-1}$, thereby indicating the elevated content of polymer carbon backbones and reduced presence of carboxyl groups on the DNR hydrogel surface. We further quantitatively characterized the surface chemical compositions of the hydrogels by X-ray photoelectron spectroscopy (*XPS*). The *XPS* spectra showed the presence of the same elements for both hydrogels but a higher carbon content for the DNR hydrogel (Supplementary Fig. 10b). We next deconvoluted the high-resolution *XPS* spectra (C 1 *s* and O 1 *s*, Fig. 1c and Supplementary Fig. 11) and determined the element content distributions on the hydrogel surface (Fig. 1d). The DNR hydrogel had a higher content of polymer carbon backbones (60 atom% versus 50 atom% of conventional hydrogel) and fewer carboxyl groups (40 atom% versus 50 atom%), which are considered hydrophobic segments and hydrophilic segments, respectively. The significant difference in the contents of elements and polymeric structures demonstrates the occurrence of network reconfiguration at the DNR hydrogel surface and subsequent development of surface hydrophobicity. We further characterized the *XPS* depth profile of the DNR

hydrogel (from the top surface to a 60 nm depth with a step of ~20 nm) by argon cluster beam etching (Supplementary Fig. 12a, b). The results showed a high carbon content (74 atom%) on the surface of the DNR hydrogel and then a gradual transition with increasing depth to a steady state (~69 atom%), similar to the value of the conventional hydrogel (68 atom%) (Supplementary Fig. 12c). The element distribution of different polymer segments at various depths was also determined by deconvolution of the high-resolution XPS spectra (Supplementary Fig. 12d, e). Moreover, the XPS depth profile of conventional hydrogel exhibits almost identical element contents across the top surface layer, drastically different from that of the DNR hydrogel surface layer (Supplementary Fig. 13). These results demonstrate that silicone chain-induced network reconfiguration only occurred at the surface of the DNR hydrogel (approximately tens of nanometers) with minimal effect on the inner hydrogel networks.

We next showed that the DNR-mediated hydrogel surface regulation does not affect hydrogel bulk properties (Supplementary Fig. 14). Rheological frequency sweep analysis revealed almost identical mechanical properties for the conventional and DNR hydrogels (Supplementary Fig. 14a). The optical photograph showed high transparency for both hydrogels (Supplementary Fig. 14b). The two hydrogels showed similar trends of swelling or drying despite the slower swelling/drying rates for the DNR hydrogel at the beginning due to its hydrophobic surface (Supplementary Fig. 14c–e).

### Dynamics of substrate-grafted polymeric chains regulates the hydrogel surface network reconfiguration

To verify our hypothesis that the structural dynamics of the grafted silicone chains is essential to the hydrogel surface network reconfiguration, we first adjusted the crosslinking degree of the silicone chains by adding 0 wt%, 50 wt%, and 100 wt% methyltrimethoxysilane (MTMS) as a crosslinker during grafting (Fig. 2a and Supplementary Fig. 15). The atomic force microscope (AFM) images showed increasing aggregation of silicone chains with increasing crosslinking, which therefore led to decreasing structural dynamics of the grafted silicone chains[31]. The surface hydrophobicity of the DNR hydrogel dramatically decreased with increasing crosslinking of the silicone chains grafted on the mold surface (Fig. 2b). Increasing the amount of crosslinker only slightly affected the surface wettability of DNR molds, i.e., ~10° reduction in WCA (Supplementary Fig. 16), which is significantly less than the drastic decrease in the WCA of DNR hydrogel. These findings indicate that the dynamics of the grafted silicone chains rather than the mold hydrophobicity is more likely to be the major contributing factor to account for such sharp change in DNR hydrogel surface wettability. To elucidate the underlying chemical mechanism of the effect of silicone chain crosslinking degree on surface properties of DNR hydrogels, we performed XPS analysis on the hydrogels prepared by DNR molds with various crosslinking degree of silicone chain (Supplementary Fig. 17). The carbon content in the hydrogel surface declined from 74 atom% to 69 atom% when the MTMS content increased from 0 wt% to 50 wt% (Supplementary Fig. 17a, b), which is very close to the value of the conventional hydrogel (68 atom%). Moreover, the gradual decrease of hydrophobic segments in the hydrogel surface demonstrates the weakened reorganization of hydrogel surface network (Supplementary Fig. 17c, d). These results corroborate the inhibition of the DNR effect by the increased silicone chain crosslinking degree on the DNR mold, thus impacting the surface properties of DNR hydrogels.

We next adjusted the chain length by altering the condensation time of dimethyldimethoxysilane (DMDMS) used for grafting silicone chains (Fig. 2c and Supplementary Fig. 18). We tested the thickness of the polymeric coating made of silicone chains by ellipsometry (Fig. 2d)[32]. Silicone chains grafted with a condensation time of 3 h showed the maximal length, and the DNR hydrogel prepared on the mold surface demonstrated the highest hydrophobicity (Fig. 2e). Meanwhile, the

analysis of the frictional property of the mold surface indicated that the longer silicone chains possessed higher structural mobility and dynamics (Supplementary Fig. 19), thereby indicating the key role of silicone chain dynamics in inducing hydrogel surface network reconfiguration.

We further tuned the chain density of silicone by varying the DMDMS concentration, from 1 wt% to 20 wt%, during grafting to manipulate the interface dynamics of the DNR mold (Fig. 2f). We determined the surface density of silicone chains by AFM topographical imaging (Fig. 2g and Supplementary Fig. 20) by quantifying the fraction of mold surface occupied by silicone (gray color) based on the color difference (Supplementary Discussion 2). The processed AFM images showed that the area fraction of silicone chains on the DNR mold increased from 79% to 96% with increasing DMDMS concentrations, suggesting a higher grafting density of silicone chains. We further verified this increasing density of silicone chains by XPS analysis (Supplementary Fig. 21). A relatively low chain density with an optimal combination of desired dynamics and a sufficient quantity of silicone chains achieved with 5 wt% DMDMS resulted in the most significant network reconfiguration and hydrophobicity of the hydrogel surface, whereas a higher chain density (10–20 wt% DMDMS), therefore lower chain dynamics, on the mold surface led to decreased hydrogel surface hydrophobicity (Fig. 2h). These results again established the causal correlation between the structural dynamics of grafted hydrophobic silicone chains and the network reconfiguration-associated hydrophobicity of the DNR hydrogel surface. From a holistic perspective, we conducted a thorough statistical comparison over the WCA of DNR hydrogels prepared by the molds grafted with silicone chains of various lengths and densities (Supplementary Fig. 22), and the results are consistent with our previous conclusions. In addition, we reason that the short length and high density of silicone chains on DNR molds could compromise the DNR effect in hydrogel surface due to reduced chain dynamics.

We investigated the influences of the parameters of hydrogel polymer networks, including the crosslinking density and content of polymer, on network reconfiguration at the hydrogel surface by controlling the content of crosslinker and monomer, respectively (Supplementary Fig. 23). The DNR hydrogel with higher crosslinking density exhibited lower surface hydrophobicity compared with the hydrogels with lower crosslinking density (Supplementary Fig. 23a). This suggests that the high hydrogel crosslinking density can hamper the hydrogel surface network reconfiguration due to reduced network dynamics. In addition, the hydrogel polymer content should be controlled at an appropriate level (20–30 wt% monomer concentration) because high polymer contents also increase the stiffness and reduce the dynamics of hydrogel networks (Supplementary Fig. 23b), thereby limiting the extent of DNR. Moreover, a low polymer content (10 wt% monomer concentration) also limits the DNR effect, possibly due to the increased water content and hydrophilicity of hydrogel surface.

We used two other common polar solvents, ethanol and dimethylsulfoxide (DMSO), to prepare PAA gels by the DNR strategy (Supplementary Fig. 24). The results show that the gels made by ethanol and DMSO are more hydrophilic than the hydrogel prepared using water, with the WCA of ~80° and ~50°, respectively. This could be attributed to the higher affinity of the organic solvents to the hydrophobic DNR mold than to the polymers, which could weaken the DNR effect of gel networks. We also prepared the DNR hydrogels with different thicknesses including 0.4 mm and 1 mm. The hydrogel bulk properties, including rheological and swelling properties, were not significantly changed (Supplementary Fig. 25).

### Molecular dynamics (MD) simulations demonstrate the DNR effect

In order to thoroughly explore the structural and dynamic details at the hydrogel–mold interface, we next conducted extensive all-atom MD simulations. Specifically, we first probed the driving forces that

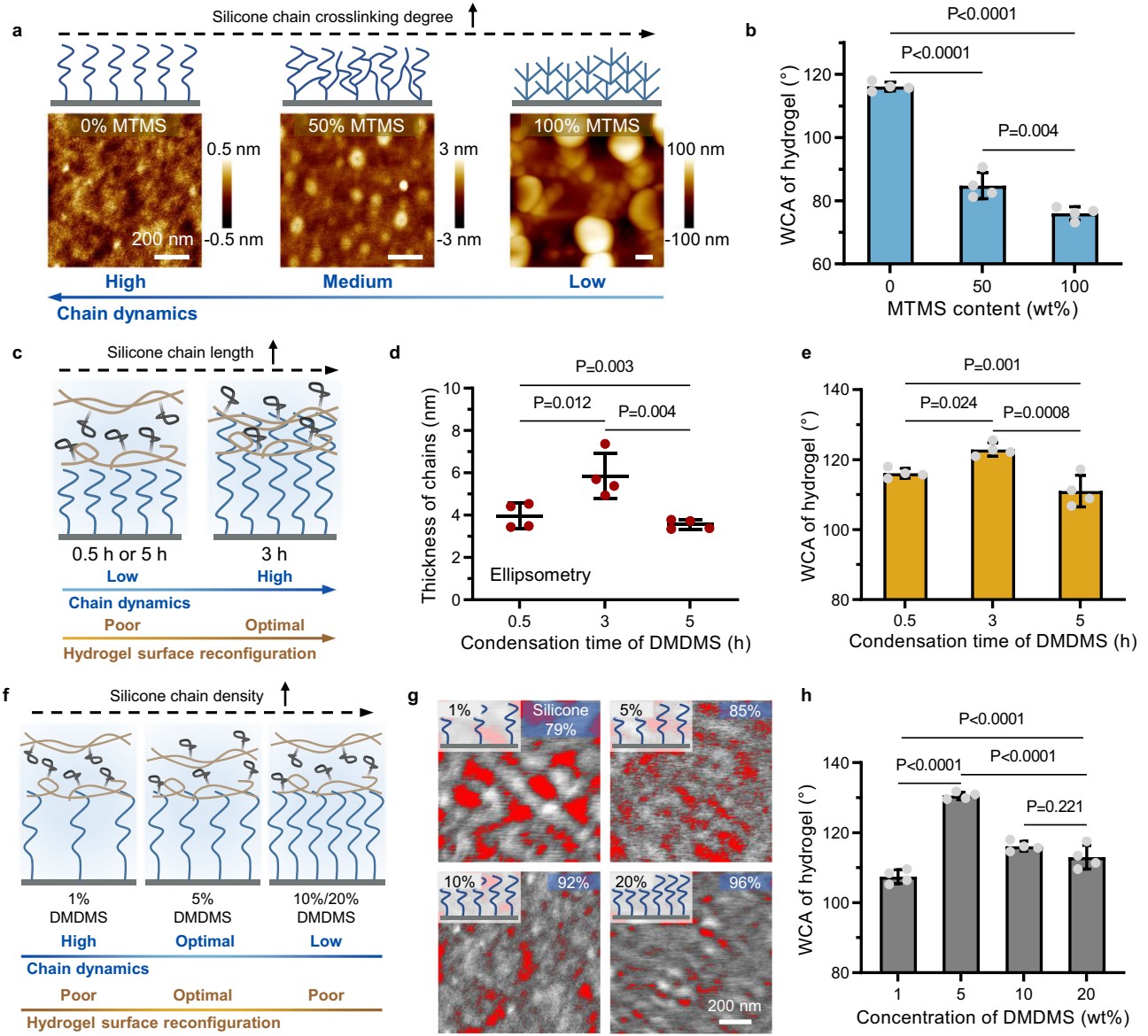

**Fig. 2 | The structural dynamics of silicone chains grafted on the mold surface is essential to hydrogel surface network reconfiguration. a** Schematic illustration and corresponding atomic force microscopy (AFM) images of the DNR mold surface grafted with silicone chains of different crosslinking degrees prepared by adjusting the amount of crosslinker methyltrimethoxysilane (MTMS). The scale bar is 200 nm. Three times each experiment was repeated independently with similar results. **b** Water contact angle (WCA) of DNR hydrogels prepared by DNR molds with different crosslinking degrees of silicone chain. **c** Schematic illustration of the hydrogel surface network interacting with grafted silicone chains of varying chain lengths obtained by adjusting the condensation time of dimethyldimethoxysilane (DMDMS). **d** Thickness of the silicone chain coating with different chain lengths determined by ellipsometry. **e** WCA of hydrogels prepared by DNR molds with different lengths of grafted silicone chains. **f** Schematic illustration of the interaction between hydrogel surface networks and silicone chains with varying chain density obtained by adjusting the DMDMS concentration during grafting. **g** Processed AFM images showing the area of the silicone chain (gray) grafted on DNR molds with different chain densities. Three times each experiment was repeated independently with similar results. **h** WCA of hydrogels prepared by DNR molds with different densities of silicone chains. Values in (**b**), (**d**), (**e**), and (**h**) are shown as the mean ± SD; $n = 4$ independent samples. Statistical analyses were performed by using ordinary one-way analysis of variance (ANOVA) with Tukey's post hoc test. $P$ values less than 0.05 were considered statistically significant differences among the compared groups.

induced the surface network reconfiguration of the DNR hydrogel. Initiating our simulations from the same PAA solution consisting of 200 PAA chains, each with 25 repeat units, we examined the hydrogel-formation on both the untreated and silicone chain-grafted glass molds. While the PAA chains aggregated rapidly and formed hydrogels on both molds within the microsecond simulations, the formed PAA hydrogels contacted the two molds in distinct manners (Fig. 3a). Compared to the silicone chain-grafted glass, the carboxyl groups of PAA formed more hydrogen bonds with the silanols on the untreated glass surface (Fig. 3b and Supplementary Fig. 26a). In contrast, on the silicone chain-grafted glass, the scarcity of hydrogen bond partners dictated that PAA contacted the silicone chains mainly with its hydrophobic backbone. Hindered by the surficial water adsorbed on the untreated glass surface, the PAA hydrogel network only adhered to approximately half of the untreated glass surface (~40%), whereas the PAA hydrogel network occupied a significantly higher proportion of the silicone chain-grafted surface (~60%, Fig. 3c and Supplementary Fig. 26b). Within the hydrogel network contacting areas, PAA preferred to expose its polar oxygens more to the untreated glass than to the silicone chain-grafted glass (Fig. 3d and Supplementary Fig. 26c), in

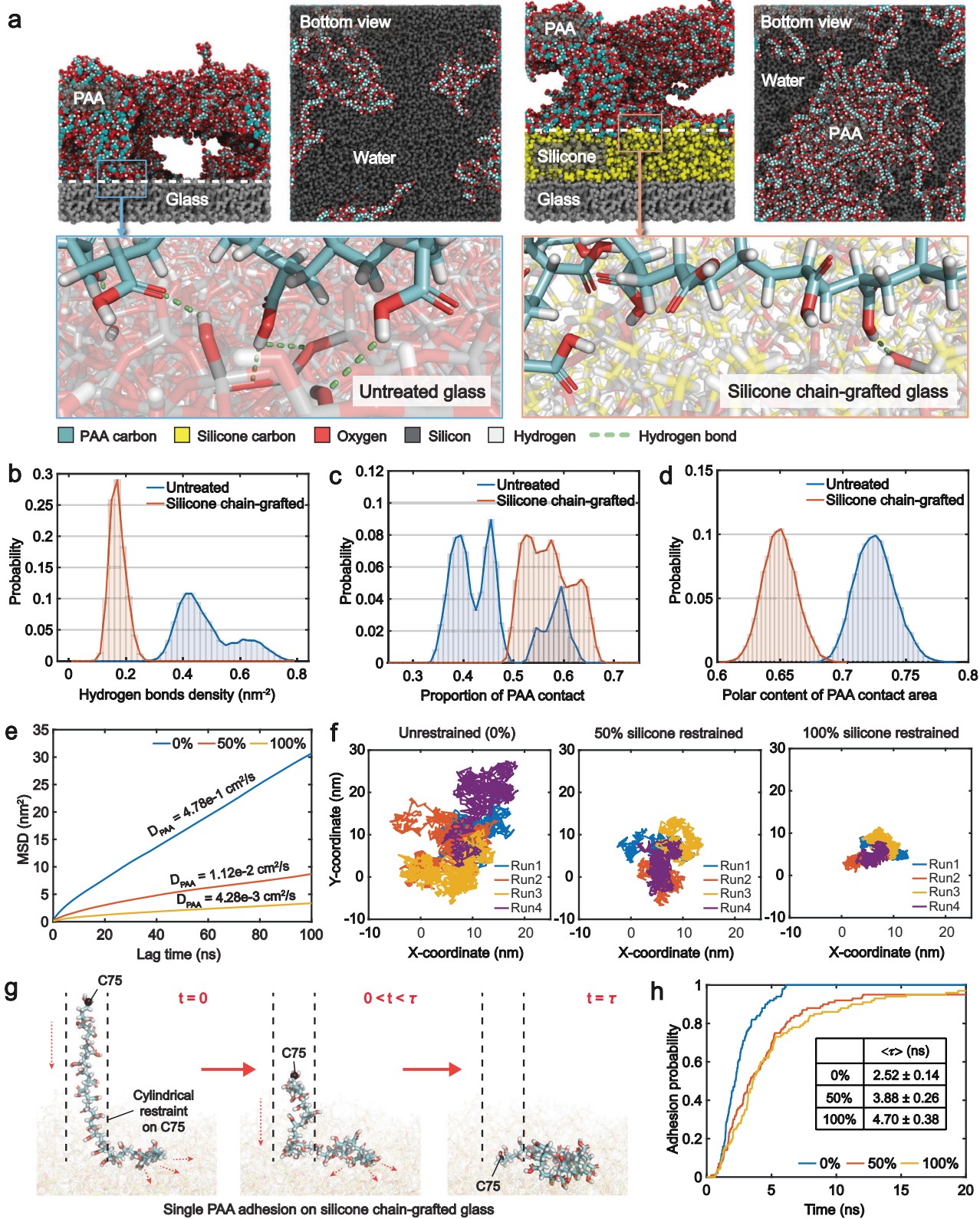

line with the greater hydrogel surface hydrophilicity obtained from the former surface. These findings indicate that when the dynamics of PAA chains are not severely hindered (as in this set of MD simulations), the dense hydrophobic interactions between silicone chains and PAA backbones can provide a strong driving force to induce efficient PAA network reconfiguration, which leads to increased hydrogel surface hydrophobicity.

To further model the dynamics of a DNR mold and probe its influence on hydrogel network reconfiguration, we then sought to quantify the diffusion of a single PAA chain adhered onto silicone chain-grafted molds with various degrees of restraints. Specifically, we position-restrained the silicon atoms in 0%, 50% or 100% of the silicone chains on the mold to mimic the effect of MTMS in crosslinking and restricting the dynamics of silicon chains (0% restraint corresponding

**Fig. 3 | MD simulations of PAA hydrogel formation and diffusive dynamics on untreated glass and silicone chain-grafted glass. a** Representative snapshots of hydrogel-formation simulations at 1 μs. The bottom views show the bottom surfaces (white dashed line) of the poly(acrylic acid) (PAA) hydrogel with the mold removed for clarity. The lower panels are magnified snapshots showing details of the interactions between PAA and molds. **b** Density of interfacial hydrogen bonds formed between the PAA hydrogel and molds. **c** Proportion of the mold surface area in contact with the PAA hydrogel surface network. **d** Proportion of polar contents (carboxylate oxygen) in the surface network of PAA hydrogels contacting the different mold surfaces. **e** Mean squared displacement (MSD) of a single PAA chain diffusing on the silicone chain-grafted mold surface with 0%/50%/100% silicone chains restrained. Diffusion coefficients of PAA were obtained from fitting of

the MSD data. **f** Trace of the C75 atom of PAA in the four 1-μs surface-diffusion simulation replicas. **g** Schematic snapshots of the adhesion-diffusion simulations for a single PAA chain on the silicone chain-grafted molds with 0%/50%/100% silicone chains restrained. The terminal carbon of PAA at its free end, C75, is restrained by a cylindrical wall potential. The adhesion time, τ, is defined as the minimum time required for the complete adhesion of PAA. **h** Adhesion probability of PAA calculated from 100 runs of 20-ns adhesion-diffusion simulations. Mean adhesion time of PAA ($\langle \tau \rangle$) ± the standard error of the mean, is given in the inserted table; $P < 0.0001$ (between the 0% and 50% data group); $P = 0.080$ (between the 50% and 100% data group); $P < 0.0001$ (between the 0% and 100% data group). Statistical analyses were performed by using two-tailed Student's $t$ test. $P$ values less than 0.05 were considered statistically significant differences among the compared groups.

to 0% MTMS), respectively. The adhered PAA chain was allowed to move freely on all mold surfaces, and its motion was recorded during four replicas of 1-μs simulations. As shown in Fig. 3e, the diffusion coefficients of the PAA chain decreased by two orders of magnitude as the proportion of restrained silicone chains increased from 0% to 100%. Over the same time span of 1 μs, the PAA chain exhibited a significantly greater sliding motion along the unrestrained mold surface (0%) compared to the restrained ones (50% and 100%) (Fig. 3f and Supplementary Movie 3). These results clearly demonstrate that the diffusion of PAA on the mold surface is significantly hindered by the reduced silicone chain dynamics.

Compared to MD simulations with a limited system size, PAA chains in a bulk hydrogel may experience much more severely hindered dynamics. We therefore developed an adhesion-diffusion model to explore the role of silicone chain dynamics in facilitating (or hindering) the reorganization of PAA with restricted dynamics. Specifically, for a single PAA chain partially adhered to a silicone chain-grafted surface, we restrained the terminal carbon of its free end using a cylindrical wall potential (Fig. 3g). That is, this terminal carbon was allowed to move freely in the vertical direction but was restricted to move within the cylinder horizontally, the latter of which mimics, at least partially, the steric hindrance posed by neighboring chains in a bulk hydrogel. This model enabled us to measure the rate at which a PAA chain reorganized its conformation driven by the favorable hydrophobic interactions with the silicon chain-grafted surface while it was subjected to restrictions on its dynamics. For each of the three silicon-chain grafted molds mentioned earlier, we conducted 100 runs of 20-ns simulations to monitor the time required by the PAA chain to achieve complete adhesion (τ). In all 100 runs, the PAA chains completely adhered to the 0%-restrained mold within just 6 ns (Fig. 3h). However, even at the end of the 20-ns simulations, a small percentage of PAA chains failed to completely adhere to the restrained mold surfaces (5% in the 50%-restrained and 3% in the 100%-restrained systems) (Fig. 3h). The mean adhesion time $\langle \tau \rangle$, obtained after excluding the small number of runs where PAA adhesion was incomplete, was found to be only 2.52 ns for the 0%-restrained mold, in contrast to the significantly longer 3.88 ns and 4.70 ns for the 50%- and 100%-restrained molds, respectively (Fig. 3h). These simulation results clearly demonstrate the hindered structural reconfiguration of PAA on those DNR molds with reduced dynamics. Collectively, MD simulations reveal molecular details of how dynamics of the silicone chains can dictate the structural reorganization of PAA on mold surfaces, thereby explaining the contribution of interface dynamics to the network reconfiguration-associated hydrophobicity of the DNR hydrogel surface.

## DNR hydrogel with enhanced bioadhesion enables effective transdermal delivery

The adhesion of hydrogels to biological tissues is promising for their diverse biomedical applications but still challenging due to the existence of interfacial water[11,12], and the complicated design of hydrogels is usually required to address this problem. We next showed that the

bioadhesion of hydrogels can be significantly enhanced by using our DNR strategy for hydrogel preparation (Fig. 4). We measured the adhesion strength of conventional and DNR hydrogels (with the same component, PAA) on dry/wet porcine skins by the pull-off test (Fig. 4a and Supplementary Fig. 27a). The two hydrogels showed almost the same adhesion strength of ~37 kPa on dry tissue surfaces (Supplementary Fig. 27b). In contrast, the DNR hydrogel demonstrated much higher adhesion strength on wet tissue surfaces (~30 kPa, comparable to its adhesion on dry tissue, Fig. 4B), which was almost 3 times that of the conventional hydrogel (~11 kPa). We further measured the interfacial toughness of hydrogels on wet porcine skin by the T-peeling test (Fig. 4c). The DNR hydrogel showed significantly higher bioadhesion with an interfacial toughness of ~500 J m$^{-2}$ compared with the conventional hydrogel (~300 J m$^{-2}$, Fig. 4d). We also demonstrated the robust wet adhesion of the DNR hydrogel on a porcine heart (Fig. 4e and Supplementary Movie 4). Compared to the easily detachable conventional hydrogel (Supplementary Movie 5), the DNR hydrogel was much harder to detach and deformed the tissue surface during the peeling process. We attribute the robust bioadhesion of the DNR hydrogel to its reconfigured hydrophobic surface network, which could repel the interfacial water to enhance close contact with tissues. We verified the repelling of interfacial water by the DNR hydrogel on porcine skin by observing sodium fluorescein aqueous solution added to the hydrogel–tissue interface by fluorescence microscopy (Supplementary Fig. 28). The hydrogel properties are comparable after the loading of fluorescein molecules (Supplementary Fig. 29). For the conventional hydrogel, a significant amount of the fluorescein solution remained at the interface and then gradually diffused into the skin with a penetration depth of ~275 μm (Fig. 4f, g). In contrast, the interfacial fluorescein solution was effectively repelled by the hydrophobic DNR hydrogel surface, and minimal fluorescein diffusion into the tissue was observed. The adhesion mechanism of our DNR hydrogel is based on the physical repelling of interfacial water by the hydrophobic hydrogel surface network and therefore enhancing surface contact between the hydrogel network and substrate, while the adhesion of phenolic hydrogels is based on the chemical bonding with the substrates. Both strategies of fabricating the bioadhesive hydrogels have its own advantages and limitations. Compared with the chemical bonding-based strategy, our DNR strategy may demonstrate a series of advantages including free of intricate chemical synthesis, ease of implementation, and wide applicability. Nevertheless, there are also limitations for the DNR hydrogels, such as the matrix polymer-dependent adhesion performance, susceptibility to the surface wettability transition, and possible batch-to-batch variations because of the repeated use of DNR molds.

We also showed that the DNR hydrogels with adaptive surface wettability, a gradual transition from hydrophobicity to hydrophilicity when exposed to water over time (Supplementary Fig. 30), enhanced the transdermal delivery of loaded cargo molecules. We used sodium fluorescein as a model molecule to observe the transdermal delivery processes by DNR and conventional hydrogels on porcine skins (Supplementary Fig. 31). Due to the initial surface hydrophobicity, DNR

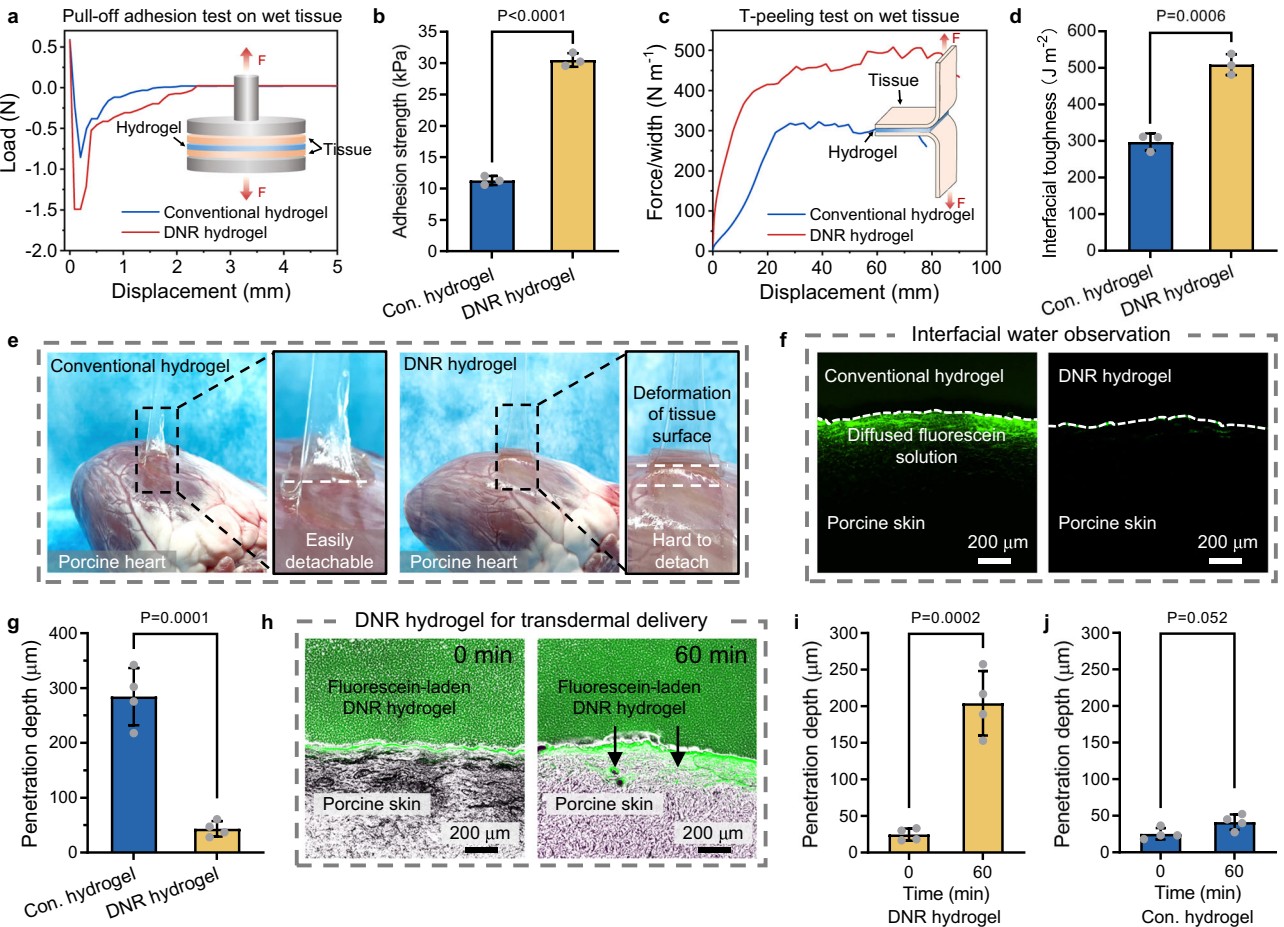

**Fig. 4 | DNR hydrogels exhibit enhanced wet bioadhesion and transdermal delivery. a** Force curves of the pull-off tests of conventional and DNR hydrogels on wet porcine skin. Poly(acrylic acid) (PAA) hydrogels prepared by the conventional method or the DNR strategy were used. **b** Adhesion strength of conventional (abbreviated as Con.) and DNR hydrogels on wet porcine skins, which was calculated by dividing peak pull-off forces by the contact area between hydrogels and tissue. **c** Force−displacement curves of hydrogel−tissue (porcine skin) hybrids from the T-peeling test. **d** Adhesion toughness of conventional and DNR hydrogels adhered on wet porcine skin. **e** Images of hydrogels adhesion on a porcine heart where the deformation of the tissue surface can be clearly observed when peeling the DNR hydrogel. **f** Fluorescence microscopy images of cryosectioned porcine skins demonstrating the diffusion of interfacial sodium fluorescein solution into

skin tissue from the hydrogel−tissue interfaces. **g** The average penetration depth of fluorescein solution for conventional and DNR hydrogels determined from the fluorescence images. **h** The merge of bright field and fluorescence images of cryosectioned porcine skins demonstrating the transdermal delivery of fluorescein into skin tissue by the DNR hydrogel at different time points. The average penetration depth of transdermal delivery of fluorescein molecules by (**i**) DNR and (**j**) conventional hydrogels at different time points determined from the fluorescence images. Values in (**b**) and (**d**) are shown as the mean ± SD, $n = 3$ independent samples; values in (**g**) and (**i, j**) are shown as the mean ± SD, $n = 4$ independent samples. Statistical analyses were performed by using two-tailed Student's $t$ test. $P$ values less than 0.05 were considered statistically significant differences among the compared groups.

hydrogels can repel interfacial water at the beginning and then promote the delivery of fluorescein molecules into skin tissue as the hydrogel surface gradually turned hydrophilic over time (Fig. 4h, i). In contrast, conventional hydrogels would be swollen by the interfacial water and the diffusion of fluorescein molecules was hindered by the swollen hydrogel interface (Fig. 4j and Supplementary Fig. 32). We studied the drug release profile of the DNR hydrogel under physiologically relevant conditions by using sodium fluorescein as the model drug (Supplementary Fig. 33). The DNR hydrogel demonstrates a burst release with 1 h and a subsequent steady release up to 20 h with the cumulative release rate of ~18%.

We examined the biocompatibility of DNR hydrogel by analyzing its cytotoxicity and inflammatory response (Supplementary Fig. 34). The results of MTT assay and live/dead staining showed that the majority of the L929 cells cultured with the DNR hydrogel extracts remained viable as also found in the control group (Supplementary Fig. 34a, b). Furthermore, hematoxylin-eosin (H&E) staining showed that the amount of inflammatory cells surrounding the DNR hydrogel implanted subcutaneously in the mice was similar as that in the control

group (Supplementary Fig. 34c), indicating the good biocompatibility of DNR hydrogel. Moreover, the blood biochemical indictors of mice receiving DNR hydrogel implantation, including alanine aminotransferase, aspartate aminotransferase, blood urea nitrogen, and creatinine, showed no significant difference compared with that of the control group (Supplementary Fig. 34d), indicating the low organ toxicity of DNR hydrogel.

Herein, we reported a simple strategy to regulate hydrogel surface wettability and bioadhesion without changing hydrogel chemical composition and bulk properties based on interface DNR. We showed that structurally dynamic silicone chains grafted on the mold surfaces can induce the conformational reconfiguration of hydrogel surface networks by interchain interactions, resulting in an elevated content of hydrophobic structures and a reduced presence of hydrophilic components in the hydrogel surface networks. The increased surface hydrophobicity led to the significant enhancement of hydrogel bioadhesion. Moreover, the DNR hydrogels showed enhanced transdermal delivery by repelling interfacial water with the initial hydrophobic surface and subsequently promoted cargo delivery as the

hydrogel surface gradually turned hydrophilic. The wide applicability of our strategy may provide a platform for customization of hydrogel surface properties to meet the needs of broad applications ranging from tissue adhesives and drug delivery to hydrogel implants and devices.

## Methods

### Materials
Dimethyldimethoxysilane (DMDMS), MTMS, methacrylic acid (MAA), *N,N'*-methylenebisacrylamide (MBAA), ammonium persulfate, and sodium fluorescein were purchased from J&K Scientific, China. Acrylamide (AAm), acrylic acid (AA), and Irgacure 2959 were obtained from Tokyo Chemical Industry, Japan. Silicone oil (PMX-200, ~500 mPa•s) was purchased from Aladdin, China. The chemicals are in analytical grade. The monomers AA and MAA were purified before use through an alkaline aluminum oxide chromatography column to remove the inhibitor. Other chemicals were used without further purification. Deionized water was used throughout the experiments. Various substrates, including glass slides, PMMA, PTFE, silicone and aluminum sheets, and silicon wafers, are commercially available.

### Preparation of the silicone chain-grafted mold (DNR mold)
Silicone chains were grafted to the mold surface via a silane hydrolysis–condensation method[32]. Without other instructions, the standard preparation procedures were as follows. A reactive solution containing 10 wt% DMDMS and 1 wt% sulfuric acid in isopropanol was first prepared. Oxygen plasma-cleaned substrates were submerged in the reactive solution for 10 s and withdrawn gradually. Excess liquid was drained via brief contact with filter paper. The mold surface was dried for 30 min to experience the condensation of DMDMS at room temperature (23 °C, 60% relative humidity), after which water, ethanol, and dichloromethane were used in sequence to rinse the molds. Because of its easy surface modification with hydrophobic chains, wide applicability, and low cost, glass substrates were used throughout the work unless otherwise specified.

For regulation of the crosslinking degree, chain length, and chain density, various amounts of MTMS (the ratio to DMDMS; 0 wt%, 50 wt%, and 100 wt%), various condensation time of DMDMS (0.5 h, 3 h, and 5 h), and various concentrations of DMDMS (1 wt%, 5 wt%, 10 wt%, and 20 wt%) were used.

### Preparation of conventional and DNR hydrogels
The untreated molds and DNR molds were used to prepare conventional and DNR hydrogels, respectively. Untreated molds were ultrasonically cleaned by ethanol before use and without changing their surface chemistry. An aqueous precursor with monomer (AA, 25 wt% in all experiments if there are no further descriptions), 0.06 wt% crosslinker (MBAA, weight ratio to monomer), and 0.15 wt% initiator (Irgacure 2959 or ammonium persulfate) was prepared. The precursor was poured into the corresponding mold with a 1-mm thick PTFE spacer and covered with another untreated or DNR mold substrate. The molds were clamped by binder clips and placed under UV light (365 nm, 1.5 mW cm$^{-2}$, 1 h) or in an oven (60 °C, 2 h) for photo/thermal polymerization. The conventional and DNR hydrogels were obtained by removing the molds. To remove any physically adsorbed residues, the hydrogels were immersed into excess deionized water and dried the apparent moisture under ambient conditions before the measurement of hydrogel surface properties.

For the preparation of PMAA and PAAm hydrogels, the precursor contained 25 wt% MAA and 20 wt% AAm, respectively. The weight ratio of the crosslinker and initiator is the same as that of the AA precursor. For preparation of PAA/gelatin and PAAm/alginate double-network hydrogels, 10 wt% gelatin (type A - 300 g bloom from porcine skin, Vetec) was added to the AA precursor, and 2 wt% sodium alginate (S100128, Aladdin) was added to the AAm precursor.

### Contact angle measurement
The WCAs of the substrates and hydrogels were measured using a contact angle meter (DSA25, Kruss). The WCA measurements were conducted with a 5-μL water droplet at room temperature (23 °C, 60% relative humidity). For the WCA measurement on hydrogels, the water drop was pipetted to the hydrogel surface, and the WCA was recorded after complete contact between the water drop and hydrogel. The recording process was continued for 5 min or 30 min to observe the WCA evolution on the hydrogel surface.

### Surface chemistry characterization
The hydrogels were immediately frozen in liquid nitrogen after preparation and freeze-dried to immobilize the surface chemistry for characterization. Freeze-dried samples were stored in a vacuum desiccator before characterization. ATR-FTIR spectroscopy was carried out to record the chemical signals of the hydrogel surface via an FTIR spectrometer (Nicolet iS50, Thermo Fisher) equipped with an ATR accessory. Transmission spectra were recorded at room temperature in the range from 4000 to 400 cm$^{-1}$ at a resolution of 0.5 cm$^{-1}$. *XPS* was used to quantitatively analyze the surface chemical composition and was performed with an X-ray photoelectron spectrometer (Axis Supra+, Kratos). The *XPS* spectra were deconvoluted using commercially available software (XPSPEAK) and atomic ratios were determined by integration of the relevant photoelectron peaks. To measure the *XPS* depth profile, the hydrogel surface was etched by an argon cluster beam (4 keV). The time of the cluster etching steps was controlled by a reference to achieve a depth resolution of ~3 nm. The FTIR and *XPS* measurements were conducted based on three independent samples with two measuring points on each sample.

### Rheological test
Rheological tests were performed on a hybrid rheometer (Discovery HR-20, TA Instruments). The frequency sweep of the hydrogel (diameter: 8 mm, thickness: 1 mm) was conducted in the range from 0.1 to 100 rad s$^{-1}$ at a strain of 0.1% and temperature of 25 °C with a normal force of ~0.01 N.

### Swelling and water loss ratio measurements
The swelling ratio of the hydrogel was measured through an equilibrium swelling experiment. A cuboid-shaped hydrogel patch (10 × 10 × 1 mm) was used for the swelling test and initially weighed as $W_o$. The hydrogel was then immersed in excess water at room temperature. At regular intervals of 30 min, the swelling hydrogel was taken out, and superficial water was removed using filter paper. The weight of the swelling hydrogel was recorded as $W_s$, and the swelling ratio ($q$) was determined by Eq. (1).

$$q = (W_s - W_o)/W_o \qquad (1)$$

The water loss ratio of the hydrogel was measured by exposing a cuboid-shaped hydrogel patch (10 × 10 × 1 mm) to the ambient environment (23 °C, 60% relative humidity). The initial weight was recorded as $W_o$. At regular intervals of 1 h, the weight of the hydrogel was recorded as $W_l$, and the water loss ratio ($Q$) was determined by Eq. (2).

$$Q = (W_o - W_l)/W_o \times 100 \qquad (2)$$

### Surface topography characterization
Atomic force microscopy (AFM) was used to characterize the surface topography of silane-treated glass slides. AFM images were acquired with a scanning probe microscope (Multimode 8, Bruker) in tapping mode. Dry samples were attached onto the sample stage with double-sided carbon tape. The probe lightly taps on the sample surface to record the surface topography. The bright regions with higher height

correspond to regions with abundant silicone chains, and the dark regions with low height correspond to regions with relatively less or no silicone chains.

## Ellipsometric thickness measurement

Ellipsometric thicknesses were measured using a Stokes ellipsometer (L116S300, Gaertner Scientific) at a 70° incident angle from the normal to the plane. The light source is a He-Ne laser ($\lambda = 632.8$ nm). The thickness was calculated using LGEMP software provided by the manufacturer. The measurement error is within 1 angstrom, as specified by the manufacturer. Thickness values were measured after silicone chain-grafted silicon wafers were rinsed thoroughly and dried in a vacuum oven at 60 °C for 2 h. Each value reported is the average of values determined at six different positions on each sample. The error bars for the thickness are the standard deviation (SD) from six independent measurements.

## Friction measurement

Friction test was conducted on a hybrid rheometer (Discovery HR-20, TA Instruments) to measure the frictional coefficient of the glass-based DNR mold. The mold samples ($20 \times 20$ mm) were attached to the sample stage by double-sided tape in a parallel plate configuration. A disposable aluminum fixture ($R = 10$ mm) was lowered against the bottom sample until a load value of -0.01 N. The samples were tested by rotating two revolutions in each direction at a constant sliding velocity of 0.03 rad s$^{-1}$. During each test, torque ($\tau$) and normal force ($N$) were measured. Instantaneous measurements of $\mu_k$, the kinetic friction coefficient, were determined by Eq. (3).

$$\mu_k = \tau / (R \times N) \quad (3)$$

The values of instantaneous $\mu_k$ were averaged over the second revolution in each direction to produce an average $\mu_k$ that was used for comparison[33].

## Molecular dynamics simulations

Molecular dynamics (MD) simulations were conducted at all-atom resolution by GROMACS-2021.4[34]. PAA chains with 25 repeating units and -OH-terminated silicone (polydimethylsiloxane, PDMS) chains with 28 repeating units were modeled using the Avogadro program[35]. PAA was parameterized using the CGenFF server[36] while silicate glass was parameterized by the INTERFACE force field (IFF)[37], and the parameters of PDMS in the silicone chain-grafted glass model were adapted from Smith et al. [38]. All remaining parameters were obtained from the CHARMM36[39] and CHARMM General force field (CGenFF)[40,41]. The neutral $Q_3$ amorphous silica model from the IFF database[37] was used to model the silicate glass mold. The silicone chain-grafted glass model was then built based on the $Q_3$ amorphous silica model. Specifically, the -OH-terminated PDMS chains were attached to randomly selected silanols on the silicate glass surface to achieve a surface density of 0.9 -OH-terminated PDMS chains per nm$^2$. All modeled structures were relaxed in water using NPT equilibrations prior to simulations. Detailed MD simulation protocols are as follows.

PAA hydrogel-formation on untreated glass and silicone chain-grafted glass molds were first simulated. PAA chains with 25 repeating units were selected as the representative structure to ensure a balanced computational performance and analogy to experimental settings. Initially, 200 PAA chains were randomly placed in a rectangular box with a lateral xy dimension of $13.5 \times 13.9$ nm$^2$ to match the size of substrates. The PAA chains were solvated with -60,000 TIP3P water to achieve a concentration of 25 wt% and then placed on top of a given substrate. Based on the experimental conditions of the monomer precursor, where 25 wt% AA was added into deionized water resulting in a solution with the pH of 1.75, we calculated the percentage of deprotonated PAA carboxyl groups ($\alpha$) based on the solvent volume ($V$) using Eq. (4),

$$\alpha = \frac{V \cdot 10^{-1.75} \cdot N_A}{n(AA)} \quad (4)$$

where $N_A$ is Avogadro's number. Therefore, 0.5% PAA carboxyl groups (25 out of 5,000 PAA repeating units) were deprotonated. Sodium ions were added to neutralize the system charge. The simulation system was periodic in the xy dimension, while a longer z dimension was used due to the presence of the substrate. To prevent PAA chains, water and ions from touching the top periodic image of the substrate, a 20.0-nm-tall vacuum slab with a lateral xy dimension matching that of the substrate was added on top of the solvated system. An extended flat-bottom potential restraint was applied to the solvents to retain a 1.6-nm water-capped buffer at the top of the systems to prevent direct contact of the solvated PAA with the vacuum slab. The hydrogel-formation system was first energy minimized and equilibrated for 1 ns in the NVT ensemble, followed by four 1-μs production simulations in the NVT ensemble. The GROMACS package was used to measure the solvent accessible surface area and intermolecular hydrogen bonds. The hydrogen bonds were defined by a cutoff radius of 0.35 nm and a cutoff angle of 30°. The contact areas of PAA and its oxygen atoms on the substrate surface were obtained by extracting the corresponding portions from the solvent accessible surface areas. The first 200 ns of each 1-μs replica was considered as equilibration (Supplementary Fig. 26) and excluded from subsequent analysis. The probability histograms for hydrogen bonds, proportion and polar content of contact areas were calculated and normalized over the last 800 ns of four replicas.

Next, MD simulations were used to investigate the impact of silicone chain dynamics on PAA chain diffusion. First, the surface-diffusion simulations were set up with the initial structure and configurations obtained from 1 μs of the hydrogel-formation simulations, where PAA hydrogel adhered to the silicone chain-grafted mold. Only one PAA chain that completely adhered to the mold was retained, while the others were removed. Position restraints with a force constant of 400 kJ·mol$^{-1}$·nm$^{-2}$ were imposed on the silicon atoms on 0%, 50%, or 100% of the silicone chains to mimic the different dynamics of silicone chains with various contents of MTMS. The system was then solvated, energy-minimized, and equilibrated in 1-ns NVT ensemble. 1-μs NVT production simulations were carried out in four replicas for each mold surface. The diffusion coefficients of PAA chains were calculated from the mean squared displacement (MSD) using the Einstein relation. Second, the adhesion-diffusion simulations were also conducted with the initial structure and configurations extracted from the hydrogel-formation simulations. Only one of the PAA chains, which partially adhered to the mold, was retained. Prior to the simulations, upward pulling was introduced to the top carbon atom, C75, with the adhered part restrained to straighten the non-adhered part of the PAA chain. Different silicone chain dynamics were then simulated by a similar position-restraint protocol to the surface-diffusion simulations. To ensure that the adhesion of PAA chain was purely driven by the diffusion of its adhered part, the C75 atom were restrained by a cylindrical wall potential with a radius of 0.5 nm and force constant of 400 kJ·mol$^{-1}$·nm$^{-2}$. The system was then subjected to solvation, energy minimization and 1-ns NVT equilibration. 20-ns NVT production simulations were carried out in 100 replicas for each mold surface. The complete adhesion of PAA was determined as the anchoring of C75 on the mold surface within a distance of <0.4 nm, and the adhesion time, $\tau$, was defined as the minimum time required to achieve complete adhesion.

In all simulations, van der Waals forces were smoothly switched off from 0.8 nm to 0.9 nm, and electrostatics were calculated using the particle mesh Ewald (PME) method with a cutoff of 0.9 nm. The systems were coupled at a temperature of 298 K with a velocity-rescaling

thermostat[42] and a pressure of 1 bar with a Berendsen barostat[43]. The LINCS algorithm[44] was applied to constrain all bonds with H-atoms. Visualizations were conducted and rendered using VMD[45] and PyMOL[46].

## Adhesion tests

The pull-off and T-peeling tests were both performed on a mechanical tester (500 N load-cell, Instron 5967) to characterize the adhesion strength and interfacial toughness between the hydrogel and tissue. For pull-off tests, porcine skin was cut to a surface area of 1 cm² and thickness of 3 mm. The backside of the porcine skin was adhered to an aluminum fixture using cyanoacrylate glue, and the epidermis of the porcine skin was covered with a hydrogel patch ($10 \times 10 \times 1$ mm) with or without interfacial water. The hydrogel-covered porcine skin was pressed against another piece of porcine skin fixed on the gripper at a pressure of 10 kPa using the mechanical tester for 5 s. The adhered tissues were then pulled by lifting the aluminum fixture, and the maximum tensile force was measured as the pull-off force. The adhesion strength was calculated by dividing the pull-off force by the surface area of the tissue.

To measure interfacial toughness, adhered porcine skin specimens with a width of 15 mm were prepared and tested using the standard T-peel test (ASTM F2256) and mechanical tester. All tests were conducted at a constant peeling speed of 50 mm min⁻¹. The measured force reached a plateau as the peeling process entered the steady state. Interfacial toughness was determined by dividing the plateau force by the width of the tissue sample. Polyethylene terephthalate film with 50-μm thickness was applied as a stiff backing for the tissues.

## Interfacial water observation

Porcine skin was wetted by sodium fluorescein aqueous solution at 0.1 mg mL⁻¹ and then a hydrogel patch was immediately applied onto the tissue with a mild pressure of ~10 kPa. The hydrogel-tissue hybrids were incubated for 10 min at room temperature and then frozen at −80 °C before cryosectioning with a cryostat microtome (CM1950, Leica). Fluorescence images of hydrogel-tissue sections were obtained using an inverted microscope (Eclipse Ti2-E, Nikon) with an excitation wavelength of 490 nm. The penetration depth of fluorescein into the porcine skin was analyzed by ImageJ.

## Transdermal delivery

The fluorescein-laden hydrogel was synthesized as described above, except that sodium fluorescein was dissolved in the AA precursor at 0.1 mg mL⁻¹. A hydrogel patch was adhered to wet porcine skin with a mild pressure ~10 kPa. The hydrogel-tissue hybrids were then incubated at 37 °C for various lengths of time (0, 30, and 60 min), and immediately frozen by liquid nitrogen before cryosectioning with a cryostat microtome (CM1950, Leica). The obtained hydrogel-tissue sections were imaged using an inverted microscope (Eclipse Ti2-E, Nikon), and the fluorescence images were analyzed by ImageJ to calculate the penetration depth of fluorescein into the porcine skin tissue.

## In vitro drug release

The DNR hydrogels laden with sodium fluorescein (0.2 mg mL⁻¹) were transferred to dialysis tubing (MWCO 2000, Thermal Scientific) and immersed in 20 ml of phosphate buffered saline (PBS) at 37 °C in an oscillating water bath. At different time points, 2 ml of solution was collected and an equal amount of PBS was refilled. The concentration of released sodium fluorescein was determined by a Shimadzu UV-3600 UV−vis spectrophotometer at 474 nm. Three independent samples were measured at each time points.

## In vitro cytocompatibility

The in vitro cytocompatibility of the DNR hydrogel was evaluated through the measurement of cell viability exposed to hydrogel extracts. Fibroblasts L929 obtained from the Cell Resource Center, Peking Union Medical College were used as a model to detect the cytotoxicity of DNR hydrogels. Briefly, 20 mg of DNR hydrogel, which is based on PAA/gelatin to mitigate the acidity of PAA, was prepared, washed with PBS buffer, and submerged into 1 mL Dulbecco's modified Eagle's medium (DMEM) for 24 h at 37 °C to obtain the extract with a concentration of 20 mg mL⁻¹. Subsequently, L929 cells were seeded on a 96-well culture plate with $2 \times 10^3$ cells per well and incubated at 37 °C in 5% CO₂ for 24 h. Afterward, the original medium was replaced by 100 μL of fresh DMEM (0 mg mL⁻¹ of hydrogel extract) or DMEM containing the hydrogel extracts with concentration of 1, 2, and 5 mg mL⁻¹, respectively. After incubation for different periods, the cytotoxicity of the hydrogels was evaluated by MTT method and live/dead assays.

## Laboratory animals

Animal experiments were performed according to the guidelines for the ethical review of laboratory animal welfare China National Standard (GB/T 35892-2018) and approved by the Animal Ethics Committee of South China University of Technology (No. 2022074). Balb/C mice (6–8 weeks old) were purchased from Hunan SJA Laboratory Animal Co., Ltd. and used for the subcutaneous implantation of DNR hydrogel (based on PAA/gelatin). The mice were housed in cages with well ventilation and light/dark cycles (12 light/12 dark), and the ambient temperature and relative humidity is 23 °C and 60%, respectively.

## In vivo biocompatibility

The mice were anesthetized with isoflurane (2–2.5%) and given buprenorphine subcutaneously (0.5 mg kg⁻¹) for pain management. Incisions in the mediodorsal skin of mice were made, and lateral subcutaneous pockets were created. Hydrogel disks (5 mm in diameter and 1 mm in thickness) were then implanted into the mice under sterile conditions. At designated time points, the mice were euthanized. The hydrogel implants were removed from the subcutaneous tissue and the skins were processed for H&E staining. Blood samples of mice were centrifuged at 11000 $g$ for 5 min, and then the plasma was used to analyze the hematological parameters by an automatic biochemical analyzer (3100, Hitachi).

## Statistical analysis

Unless otherwise specified, all data are presented as the mean ± SD via at least triplicate samples. Statistical analyses were performed by using ordinary one-way analysis of variance (ANOVA) with Tukey's post hoc test or two-tailed Student's $t$ test to compare multiple groups or two groups (GraphPad Prism 9.0), respectively. $P$ values less than 0.05 were considered statistically significant differences among the compared groups.

## Reporting summary

Further information on research design is available in the Nature Portfolio Reporting Summary linked to this article.

# Data availability

The data supporting the findings from this study are available within the Article, Supplementary Information, or Source Data file. Source data are provided with this paper. The neutral $Q_3$ amorphous silica model was obtained from the INTERFACE force field database (http://bionano.uakron.edu/the-interface-force-field/). Source data are provided with this paper.

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

## Acknowledgements

This work was supported by the National Key Research and Development Program (2022YFB3804403, L.B.), Science and Technology Innovation Project of Foshan City (1920001000025, L.B.) and the Collaborative Research Fund from the Research Grants Council of Hong Kong (Project No. C5044-21G, L.B.). B.Y. acknowledges the project funded by China Postdoctoral Science Foundation (2023M731125, B.Y.) and the support from Postdoctoral Research Fund of Guangzhou (Project No. L2230510, B.Y.).

## Author contributions

B.Y. and L.B. conceived the idea. B.Y. and B.G.Y. conducted the hydrogel preparation and characterizations, and data analysis. T.L. performed the MD simulations. S.C. and B.G.Y. carried out the animal and cell experiments. J.Z., K.Z., and P.Z. helped on the data analysis and discussion. B.Y. and T.L. wrote the paper. L.B., Y.W., and Z.W. revised the paper. L.B., Y.W., and Z.W. supervised the entire research. All authors approved the final version of the manuscript.

## Competing interests

The authors declare no competing interests.
