## [Peer Review File · Nature Communications]

Surface hydrophobization of hydrogels via interface dynamics-induced network reconfigurationREVIEWER COMMENTS

Reviewer #1 (Remarks to the Author):

This manuscript by Bian et al. reported an interesting surface hydrophobization strategy for hydrogels via interface dynamics-induced network reconfiguration. The authors showed that the grafting of hydrophobic yet flexible polymeric chains on mold substrates could significantly elevate the content of hydrophobic polymer backbones and reduce the presence of polar groups in hydrogel surface networks, thereby transforming the otherwise hydrophilic hydrogel surface into a hydrophobic surface. They also showed that the hydrogels prepared showed substantially enhanced bioadhesion and transdermal delivery. This work provides a novel strategy to obtain the hydrophobic surface of the hydrogel. Overall, this manuscript is well organized. This manuscript could be considered for publication in Nature communications. However, there are still some critical issues that should be solved.

Specific comments:

1. Since the air is most hydrophobic, how about the contact angle of the PAA hydrogel facing the air side? It is suggested the authors also comment on the impact of air on the surface properties of the hydrogels.
2. The authors showed that the content of the hydrogel might also have an impact during the DNR strategy. However, I would suggest that more concentrations, such as 2-10%, to be discussed.
3. How about the surface properties when using other polar solvents rather than water, such as ethanol or THF, during the DNR strategy?
4. Are the surface wetting results immediately obtained after removing the sample from the mold? The authors should rinse the sample immediately after the preparation with water to remove any physically adsorbed residues.
5. The authors claimed that the diffusion of PAA on the mold surface is significantly hindered by the reduced silicone chain dynamics. In fact, I would suspect the diffusion of the radicals was also hindered, and the polymerization or cross-linking at the interface is also different from the bulk. The authors should provide more details on this aspect. That means if the thickness of the gap or the hydrogels decreased, the properties of the bulk hydrogels might also changed.

Reviewer #2 (Remarks to the Author):

The research paper titled "Surface Hydrophobization of Hydrogels via Interface Dynamics-Induced Network Reconfiguration" introduces an innovative technique for tailoring the surface characteristics of hydrogels. By employing a mold composed of hydrophobic silicone chains, hydrogel monomers arrange

themselves with their hydrophobic segments interacting with the hydrophobic mold surface. Consequently, the polar groups of the monomers orient away from the mold, leading to the creation of a hydrophobic surface. The authors demonstrate that this technique can be used to create hydrophobic surfaces on three different types of hydrogels without changing their chemical composition. The overall concept is interesting and does have the potential to be very helpful in the biomaterials field as the authors are correct that changing surface chemistry without changing overall composition is an important aim. The authors have done an extensive amount of high-quality characterization to understand the system and their methods are described in enough detail to be reproducible with some minor exceptions (noted below). The conclusions appear to be mostly sound, again with some minor exceptions noted below. After addressing the concerns listed below, this paper could potentially be acceptable for publication in Nature Communications.

Major points:

1. The authors draw substantial conclusions on change in surface chemistry based on very small differences in the XPS and FTIR spectra. Although the conclusions may very well be correct, it is difficult to judge when everything contains exactly the same elements in nearly identical concentrations. To help increase credibility, the authors can mention how many places on their samples as well as how many independent samples were measured. Right now the caption for Figure 1D says "Data are shown as the mean +/- SD, N = 4", but no +/- values are actually given either there or in the text and there is no clarification of the N=4 in the text or materials and methods section.
2. The assertion made by the authors regarding the impact of hydrophobic monomer solubility and stability on hydrogel structure needs backing up either through relevant data or references from existing literature. Similarly, citations are also needed when discussing the introduction of hydrophobic monomers leading to unfavorable effects on bulk properties.
3. The manuscript overall could greatly benefit from more comparison with existing literature. As it stands, there is comparatively little discussion of what is already known about creating hydrogels in hydrophobic vs hydrophilic molds and therefore not a clear framing of the gaps that this work is filling.
4. Is there a specific reason to why glass substrates are used mainly throughout the paper instead of the other substrates mentioned? Elaboration would help the reader better understand the system.
5. In Figure S5C, it would help to clarify why there is a more significant reduction in water contact angle (WCA) within 0.5 minutes for PAAm, as compared to PAA and PMAA? Furthermore, is there a specific rationale for the absence of PAAm and PMAA in Figure S5D?
6. The authors have indicated that the water contact angle (WCA) value decreased more rapidly for the oil-coated mold in comparison to DNR. However, Figure S8C lacks the representation of the decline rate, depicting instead that the silicone oil-coated hydrogel initiated with a lower WCA and followed a similar trajectory as the DNR hydrogel. To assert a "faster" decline, it might be more suitable to employ a decline percentage on the y-axis, providing a clearer visualization of the comparative trends.

7. The authors have indicated that network reconfiguration took place exclusively at the surface of the DNR hydrogel, with minimal impact on the inner hydrogel network (Figure S12). Notably, the depth profile of the conventional hydrogel is absent. Including this data for direct comparison would significantly bolster the credibility of the claim.

8. The authors assert that mold hydrophobicity is not “essential” to the resultant surface wettability of DNR gel, as shown in Figure 2B. However, it is important to acknowledge that although the effect might not be as pronounced, the mold hydrophobicity still decreases in correlation with the water contact angle (WCA) of the hydrogel. Therefore, its potential significance cannot be entirely dismissed.

9. The authors assert that the optimal performance is achieved with a 3-hour condensation time, as indicated in Figure 2E. In addition, a 5wt% concentration of DMDMS yields the highest promotion of hydrogel hydrophobicity (Figure 2H). However, it appears that the statistical comparison performed here is between 0.5h and 5h in Figure 2E and between 1 and 20 wt% in Figure 2H. To provide a more robust analysis, the authors should conduct a statistical comparison specifically between the 3-hour condensation time and the other time points in Figure 2E, as well as between the 5 wt% concentration and other concentrations in Figure 2H. This approach would effectively demonstrate whether these specific conditions are indeed statistically superior.

10. The mention of "sodium fluorescein" should be introduced much earlier in the paper, ideally when it is first referenced. Likewise, considering that "sodium fluorescein" is solely mentioned in the "Materials" section, reiterating its presence under the "Interfacial water observation" section, where it is utilized in the experiment, would be advantageous for readers. This approach would eliminate the need for readers to constantly refer back and forth, ensuring a more streamlined understanding of the experimental details.

11. Does the incorporation of this fluorescein molecule into the hydrogel have any impact on its properties? Supporting this with past literature citations or relevant data would be crucial to validate the rationale behind using this molecule in the experiments.

12. The authors mentioned the gradual transition of the hydrogel surface towards hydrophilicity over time. It would be helpful to specify the duration required for this transformation to take place. Additionally, once the surface becomes hydrophilic, is there any potential impact on its adhesion properties? Elaborating on this aspect would provide a more comprehensive understanding of the evolving characteristics of the DNR hydrogel.

Minor points:

1. In figure S3A, what is the substrate of the DNR mold?

2. The abbreviation "Al" in Figure S3B requires clarification. Additionally, it is worth noting why materials such as PMMA and PTFE, presented in Figure S3A, are absent from Figure S3B. This omission raises

questions about the selection of materials for presentation in the context of this comparison.

3. Incorporating time stamps within Movie S1 and S2 would facilitate a more effective comparison between the two videos.

4. In Figure S10B, there appears to be a generally higher intensity for the conventional hydrogel. Assuming this difference is for visualization purposes, it would be advisable to explicitly state this clarification within the figure caption.

5. To offer readers a comprehensive perspective, it could be useful to include a conventional hydrogel attachment to a porcine heart in addition to Movie S3. This addition would provide a visual contrast for readers to better understand the differences between the two scenarios.

6. Incorporating appropriate significance markings on the graph would enhance its readability. Currently, the error bars overlap between the 50 and 100 data points in Figure 2B without any indication, which might imply lack of statistical significance, which is a similar case in Figure 2E and Figure 2H. In Figure 3H, the main text claims there is a statistical significance, it is also not presented in the caption nor on the figure. In Figure 3H, the main text asserts the presence of statistical significance; however, this information is not indicated in the caption nor on the figure itself.

7. It is advisable to include a mention of the mold substrate used for grafting hydrophobic silicone chains within the main text. Presently, readers must refer to caption of Figure S1b to acquire this information (glass mold), which could be more seamlessly integrated into the narrative.

8. For clarity, the authors are encouraged to cite only those parts of the figures that are relevant to the statement being made. For instance, while discussing the use of the DNR strategy to maintain hydrogel's chemical composition and bulk properties, citing Figure 1B instead of Figure 1 could help to save the reader time.

9. The authors should consider increasing the contrast difference in the colored figures to ensure that they remain distinguishable for readers with color blindness.

10. In Figure S9A, the inclusion of a title in Chinese could be addressed by either omitting it from the graph or providing a translation to English. This adjustment would ensure clarity and consistency in the presentation of the figure.

Reviewer #3 (Remarks to the Author):

This manuscript presents a novel gelation strategy called the 'Interface-mediated dynamic network reconfiguration (DNR)', which effectively transforms the surface hydrophilicity of hydrogels into a hydrophobic state while maintaining the chemical composition and overall characteristics of the hydrogels. The hydrogels prepared using DNR strategy exhibited substantially enhanced bioadhesion and improved transdermal delivery as compared with the hydrogels fabricated via the conventional method. The concept in this study is novel and fascinating, and the manuscript is concisely well written and provides many convincing data to support the authors' claims. This reviewer recommends minor revision

to address the following issues before publication.

1. Figure S4 shows the surface hydrophobicity of the PAA-based DNR hydrogel after 24 hours in a humid environment. It would also be beneficial to provide the pictures of the hydrogel to show any changes in the shape or structure of the hydrogel in the humid environment.
2. Figure S7 is showing fabrication of the Janus hydrogels with asymmetric wettability using the DNR strategy, which provides an interesting avenue of the study. Additional characterization of the Janus hydrogels to validate asymmetric wettability would be valuable information to the readers (e.g., WCA comparison in different side of the hydrogel).
3. Figure S13 demonstrates the bulk properties of the hydrogel. Regarding the swelling property of the hydrogels, the pictures of the hydrogels over time in swelling test would verify the maintenance of bulk property of the hydrogels in wet conditions.
4. In Figure 2B, the authors mentioned that the amount of crosslinker did not significantly affect the wettability of the DNR mold. However, the graph of Figure 2B seems to show some notable differences in the WCA of molds. The difference is not statistically significant?
5. Figure 2 provides important data since it nicely demonstrates that the structural dynamics of silicone chains grafted on the mold surface are crucial to reconfigure hydrogel surface networks. This figure indicates that silicon crosslinking degree, chain length, and chain density show different effects on gelation process of DNR hydrogels and surface properties of the resultant DNR hydrogels. Thus, this reviewer recommends the authors to elaborate further discussion on feasible underlying chemical mechanisms of these factors impacting on the surface properties of DNR hydrogels.
6. Figure 3 is showing the molecular dynamics simulations to analyze the structural and dynamic details of the PAA hydrogels. I just wonder whether other types of polymers mentioned in Figure 1B (PMAA, PAAm) exhibit similar dynamic behaviors when fabricated through DNR protocol.
7. Figure 4 nicely presents application of DNR hydrogel for wet bioadhesion and transdermal delivery. If the authors consider biomedical applications, biocompatibility of the DNR hydrogel, such as cytotoxicity and inflammation, should be investigated. Also, drug release profiles from DNR hydrogel may need to be examined under physiologically relevant conditions.
8. Please provide discussion to compare advantages and disadvantages between DNR hydrogels and well-known marine-inspired adhesive hydrogels functionalized with phenolic moieties (e.g., catechol, gallol).

Response to the Comments from the Reviewers:

We sincerely appreciate the reviewers' positive comments and the feedback provided to strengthen the manuscript. Here we provide a point-by-point response to each of the reviewers. The corresponding changes to the revised manuscript have been highlighted in yellow.

Reviewer #1:

This manuscript by Bian et al. reported an interesting surface hydrophobization strategy for hydrogels via interface dynamics-induced network reconfiguration. The authors showed that the grafting of hydrophobic yet flexible polymeric chains on mold substrates could significantly elevate the content of hydrophobic polymer backbones and reduce the presence of polar groups in hydrogel surface networks, thereby transforming the otherwise hydrophilic hydrogel surface into a hydrophobic surface. They also showed that the hydrogels prepared showed substantially enhanced bioadhesion and transdermal delivery. This work provides a novel strategy to obtain the hydrophobic surface of the hydrogel. Overall, this manuscript is well organized. This manuscript could be considered for publication in Nature communications. However, there are still some critical issues that should be solved.

Response: We highly appreciate the reviewer's constructive comments, and these suggestions are very helpful in improving our work.

1. Since the air is most hydrophobic, how about the contact angle of the PAA hydrogel facing the air side? It is suggested the authors also comment on the impact of air on the surface properties of the hydrogels.

Response: We thank the reviewer for raising the critical question. We found that it is difficult to polymerize the PAA hydrogel in the air. Since the hydrogel was prepared by radical polymerization, the oxygen will greatly inhibit the polymerization by consuming radicals, which is known as the oxygen-inhibition effect¹. Therefore, the properties of the hydrogel surface exposed to ambient air are affected by this oxygen-inhibition effect. Instead, we have prepared the hydrogel in a nitrogen atmosphere without oxygen. The water contact angle (WCA) measurement results shown below reveal no statistical difference between the surface wettability of hydrogels prepared in glass mold and in nitrogen atmosphere, indicating the negligible effect of nitrogen atmosphere on hydrogel surface properties.

Figure caption: WCA of PAA hydrogels prepared in glass mold and in N₂ atmosphere. Data are shown as the mean \pm SD; $N = 4$ independent samples.

2. The authors showed that the content of the hydrogel might also have an impact during the DNR strategy. However, I would suggest that more concentrations, such as 2-10%, to be discussed.

Response: We thank the reviewer for the suggestion. We have tried to prepare PAA hydrogel with the monomer concentration of 2 wt%, 5 wt% and 10 wt% under the same crosslink density as the hydrogel with 20–40 wt% monomer concentrations. However, it cannot form a hydrogel with only 2 wt% and 5 wt% monomers. Therefore, we studied the WCA of the hydrogel prepared by 10 wt% monomer. As shown in the revised Figure S23B, the DNR hydrogel prepared by 10 wt% monomer displays a lower WCA than that by 20–30 wt% monomers. This suggests that a lower polymer content also impacts the DNR effect, possibly due to the increased water content improving the hydrophilicity of hydrogel surface.

We have added the discussion in paragraph 1 on page 11.

3. How about the surface properties when using other polar solvents rather than water, such as ethanol or THF, during the DNR strategy?

Response: We thank the reviewer for this valuable question. We used two other common polar solvents, ethanol and dimethylsulfoxide (DMSO), to prepare PAA gels by the DNR strategy (Figure S24). The results show that the gels made by ethanol and DMSO are more hydrophilic than the hydrogel prepared using water, with the WCA of $\sim 80^\circ$ and $\sim 50^\circ$, respectively. This could be attributed to the higher affinity of the organic solvents to the hydrophobic DNR mold than to the polymers, which could weaken the DNR effect of gel networks.

We have added the discussion in paragraph 2 on page 11.

Fig. S24. WCA of PAA gels prepared using the DNR strategy with different polar solvents, including water, ethanol, and DMSO (dimethylsulfoxide). The preparation process of the gels with ethanol and DMSO is the same with the hydrogel except the change of solvent. Data are shown as the mean \pm SD; $N = 4$ independent samples.

4. Are the surface wetting results immediately obtained after removing the sample from the mold? The authors should rinse the sample immediately after the preparation with water to remove any physically adsorbed residues.

Response: We thank the reviewer for raising this important question and we are sorry for the confusion caused. To remove any physically adsorbed residues, we have immersed the hydrogels into excess deionized water and dried the apparent moisture under ambient conditions before the measurement of hydrogel surface properties.

We have added the description in paragraph 1 on page 20.

5. The authors claimed that the diffusion of PAA on the mold surface is significantly hindered by the reduced silicone chain dynamics. In fact, I would suspect the diffusion of the radicals was also hindered, and the polymerization or cross-linking at the interface is also different from the bulk. The authors should provide more details on this aspect. That means if the thickness of the gap or the hydrogels decreased, the properties of the bulk hydrogels might also change.

Response: We thank the reviewer for this valuable question. We have prepared DNR hydrogels with the thickness decreased from 1 mm to 0.4 mm. The hydrogel bulk properties, including rheological and swelling properties, were characterized (Figure S25) and not significantly changed. It should be noted that the silicone chains on the DNR mold only influence the top surface layer (tens of nanometers in thickness) of the hydrogel according to the analysis of the XPS depth profile (Figure S12). Therefore, we believe that the changes in such a thin layer of surficial hydrogel layer would not significantly affect its bulk properties even if the diffusion of radicals was affected.

We added the discussion in paragraph 2 on page 11.

Fig. S25. (A) Rheological frequency sweep and (B) swelling ratio of DNR hydrogels with different thickness. Values in B are shown as the mean \pm SD; $N = 3$ independent samples.

Reviewer #2:

The research paper titled "Surface Hydrophobization of Hydrogels via Interface Dynamics-Induced Network Reconfiguration" introduces an innovative technique for tailoring the surface characteristics of hydrogels. By employing a mold composed of hydrophobic silicone chains, hydrogel monomers arrange themselves with their hydrophobic segments interacting with the hydrophobic mold surface. Consequently, the polar groups of the monomers orient away from the mold, leading to the creation of a hydrophobic surface. The authors demonstrate that this technique can be used to create hydrophobic surfaces on three different types of hydrogels without changing their chemical composition. The overall concept is interesting and does have the potential to be very helpful in the biomaterials field as the authors are correct that changing surface chemistry without changing overall composition is an important aim. The authors have done an extensive amount of high-quality characterization to understand the system and their methods are described in enough detail to be reproducible with some minor exceptions (noted below). The conclusions appear to be mostly sound, again with some minor exceptions noted below. After addressing the concerns listed below, this paper could potentially be acceptable for publication in Nature Communications.

Response: We greatly appreciate the reviewer for the encouraging feedback on our manuscript and the constructive suggestions.

1. The authors draw substantial conclusions on change in surface chemistry based on very small differences in the XPS and FTIR spectra. Although the conclusions may very well be correct, it is difficult to judge when everything contains exactly the same elements in nearly identical concentrations. To help increase credibility, the authors can mention how many places on their samples as well as how many independent samples were measured. Right now the caption for Figure 1D says "Data are shown as the mean +/- SD, N = 4", but no +/- values are actually given either there or in the text and there is no clarification of the N=4 in the text or materials and methods section.

Response: We thank the reviewer for the reminder and we are sorry for the confusion caused. The caption of "Data are shown as the mean +/- SD, N = 4" in Figure 1 describes the multiple repetitions of WCA measurement shown in Figure 1B. The XPS and FTIR measurements were conducted based on three independent samples with two measuring points on each sample.

We have revised the caption of Figure 1D and added the description in "Surface chemistry characterization" section on page 20 for the XPS and FTIR tests.

2. The assertion made by the authors regarding the impact of hydrophobic monomer solubility and stability on hydrogel structure needs backing up either through relevant data or references from existing literature. Similarly, citations are also needed when discussing the introduction of hydrophobic monomers leading to unfavorable effects on bulk properties.

Response: We thank the reviewer for the kind reminder. We have added Reference 29 and 30 in paragraph 1 on page 6 to support our discussion. Ref. 29 shows that the hydrophobic monomers need to be emulsified in the hydrogel precursor to prepare a stable yet non-transparent hydrogel². Ref. 30 demonstrates that the addition of hydrophobic monomer would change the transparency of hydrogel and make the hydrogel more fragile, which can support our discussion³.

3. The manuscript overall could greatly benefit from more comparison with existing literature. As it stands, there is comparatively little discussion of what is already known about creating hydrogels in hydrophobic vs hydrophilic molds and therefore not a clear framing of the gaps that this work is filling.

Response: We thank the reviewer for this constructive suggestion. Previous work indicates that hydrogels prepared by a hydrophobic mold demonstrate a less crosslinked surface with lower modulus, compared to that prepared in a hydrophilic mold⁴⁻⁶. However, few prior studies have systemically investigated the impact of chain dynamics of mold surface-grafted hydrophobic polymers on the network reorganizations and wettability of hydrogel surfaces and associated mechanisms. Therefore, we believe that our work provide critical mechanistic insights to the hydrogel – mold surface interactions and associated changes in hydrogel surface properties from a molecular perspective, which is largely absent from prior literatures.

We have added the discussion in paragraph 2 on page 5.

4. Is there a specific reason to why glass substrates are used mainly throughout the paper instead of the other substrates mentioned? Elaboration would help the reader better understand the system.

Response: We thank the reviewer for the reminder. The use of glass substrates throughout the work is mainly because of its easy surface modification with hydrophobic chains, wide applicability, and low cost.

We have added the explanation in the methods section in paragraph 3 on page 19 to make the selection of mold substrates more clear.

5. In Figure S5C, it would help to clarify why there is a more significant reduction in water contact angle (WCA) within 0.5 minutes for PAAm, as compared to PAA and PMAA? Furthermore, is there a specific rationale for the absence of PAAm and PMAA in Figure S5D?

Response: We thank the reviewer for the helpful questions. The more significant reduction of WCA for PAAm hydrogel is possibly due to the lower polymer content of PAAm hydrogel (20 wt%) compared to that (25 wt%) of PAA and PMAA hydrogels, which like makes the hydrogel more permeable by the probing water droplet. On the other hand, we reason that the difference in side functional groups, i.e., amino in PAAm and carboxyl in PAA/PMAA, could also result in the varied recovery rate of the hydrogel surface network when contacting with the probing water droplet. We have

added the explanation under the caption of Figure S5.

We are sorry for the absence of WCA data for PAAm and PMAA hydrogels in Figure S5D. We have added the relevant results in the revised Figure S5D.

6. The authors have indicated that the water contact angle (WCA) value decreased more rapidly for the oil-coated mold in comparison to DNR. However, Figure S8C lacks the representation of the decline rate, depicting instead that the silicone oil-coated hydrogel initiated with a lower WCA and followed a similar trajectory as the DNR hydrogel. To assert a "faster" decline, it might be more suitable to employ a decline percentage on the y-axis, providing a clearer visualization of the comparative trends.

Response: We thank the reviewer for the reminder. We have added the decline percentage of WCA values in the revised Figure S8D. It shows that the silicone oil-coated hydrogel displays a higher decline rate than DNR hydrogel.

7. The authors have indicated that network reconfiguration took place exclusively at the surface of the DNR hydrogel, with minimal impact on the inner hydrogel network (Figure S12). Notably, the depth profile of the conventional hydrogel is absent. Including this data for direct comparison would significantly bolster the credibility of the claim.

Response: We thank the reviewer for this constructive suggestion. We have added the depth profile of the conventional hydrogel in the revised Figure S13, which shows that the conventional hydrogel almost exhibits almost identical element contents across the top surface layer, drastically different from the DNR hydrogel (Figure S12). Relevant discussion has also been added in paragraph 2 on page 7.

Fig. S13. (A) XPS depth profile of the conventional (Con.) hydrogel with depth from the top surface to ~60 nm. (B) Element contents of carbon and oxygen on the conventional hydrogel surface at various depths. The values were determined by integration of the C 1s and O 1s peak areas. (C) High-resolution XPS spectra of the conventional hydrogel depth profile. (D) Statistics of the element content distribution of polymer segments in the conventional hydrogel at various depths.

8. The authors assert that mold hydrophobicity is not “essential” to the resultant surface wettability of DNR gel, as shown in Figure 2B. However, it is important to acknowledge that although the effect might not be as pronounced, the mold hydrophobicity still decreases in correlation with the water contact angle (WCA) of the hydrogel. Therefore, its potential significance cannot be entirely dismissed.

Response: We thank the reviewer for this kind reminder. We have amended our description in paragraph 3 on page 8: “Increasing the amount of crosslinker only slightly affected the surface wettability of DNR molds, i.e., $\sim 10^\circ$ reduction in WCA (Figure S16), which is significantly less than the drastic decrease in the WCA of DNR hydrogel. These findings indicate that the dynamics of the grafted silicone chains rather than the mold hydrophobicity is more likely to be the major contributing factor to account for such sharp change in DNR hydrogel surface wettability”.

9. The authors assert that the optimal performance is achieved with a 3-hour condensation time, as indicated in Figure 2E. In addition, a 5wt% concentration of DMDMS yields the highest promotion of hydrogel hydrophobicity (Figure 2H). However, it appears that the statistical comparison performed here is between 0.5h and 5h in Figure 2E and between 1 and 20 wt% in Figure 2H. To provide a more robust analysis, the authors should conduct a statistical comparison specifically between the 3-hour condensation time and the other time points in Figure 2E, as well as between the 5 wt% concentration and other concentrations in Figure 2H. This approach would effectively demonstrate whether these specific conditions are indeed statistically superior.

Response: We thank the reviewer for the constructive suggestion. As recommended by the reviewer, we have conducted a thorough statistical comparison over the WCA of DNR hydrogels prepared by the molds with various concentration and condensation time of DMDMS (Figure S22), and the results are consistent with our previous conclusions.

We have added the results to Figure S22 and the relevant discussion in paragraph 2 on page 9.

Fig. S22. WCA of PAA hydrogels prepared by DNR molds with different lengths and densities of grafted silicone chains. The length of silicone chains was regulated by different condensation time of DMDMS from 0.5 h to 5 h, where the density was regulated by various concentration of DMDMS from 1 wt% to 20 wt%. Data are shown as the mean \pm SD; $N = 4$ independent samples; *** $P < 0.001$.

10. The mention of "sodium fluorescein" should be introduced much earlier in the paper, ideally when it is first referenced. Likewise, considering that "sodium fluorescein" is solely mentioned in the "Materials" section, reiterating its presence under the "Interfacial water observation" section, where it is utilized in the experiment, would be advantageous for readers. This approach would eliminate the need for readers to constantly refer back and forth, ensuring a more streamlined understanding of the experimental details.

Response: We thank the reviewer for the kind reminder and we are sorry for the inconvenience caused. We have added "sodium fluorescein" on page 16 and the caption of Figure 4F, where it is first referenced. "Sodium fluorescein" is also mentioned in "Interfacial water observation" section on page 24.

11. Does the incorporation of this fluorescein molecule into the hydrogel have any impact on its properties? Supporting this with past literature citations or relevant data would be crucial to validate the rationale behind using this molecule in the experiments.

Response: We thank the reviewer for the question and suggestion. We compared the appearance, surface wettability and rheological properties of the DNR hydrogel before and after incorporation of the fluorescein molecules (Figure S29). The results show that the hydrogel properties are comparable after the loading of fluorescein molecules, except the change of color.

We have added the description in paragraph 1 on page 16.

Fig. S29. Comparison of the DNR hydrogel before and after loading of sodium fluorescein molecules. (A) Photo images, (B) WCA, and (C) rheological frequency sweep of DNR hydrogel and fluorescein-laden DNR hydrogel. Values in B are shown as the mean \pm SD; $N = 4$ independent samples. Values in C are shown as the mean \pm SD; $N = 4$ independent samples.

12. The authors mentioned the gradual transition of the hydrogel surface towards hydrophilicity over time. It would be helpful to specify the duration required for this transformation to take place. Additionally, once the surface becomes hydrophilic, is there any potential impact on its adhesion properties? Elaborating on this aspect would provide a more comprehensive understanding of the evolving characteristics of the DNR hydrogel.

Response: We thank the reviewer for this considerate suggestion. We further measured the WCA evolution profile of DNR hydrogel in 80 min, which shows the full transition of the hydrogel surface wettability from hydrophobicity to hydrophilicity over time (Figure S30A). In addition, we found that this wettability transition may weaken the adhesion of DNR hydrogel to wet tissues to some extent (Figure S30B). Nevertheless, the early hydrophobicity of DNR hydrogel facilitates the bioadhesion on wet tissues.

We have added these results to Figure S30 and the discussion under the caption of Figure S30.

Fig. S30. (A) WCA evolution of the DNR hydrogel in 80 min, which shows the hydrophobicity to hydrophilicity transition of DNR hydrogel over time. (B) Adhesion toughness of the DNR hydrogel before and after turning hydrophilic.

hydrogel before and after turning hydrophilic adhered on wet porcine skin. Data are shown as the mean \pm SD; $N = 3-4$ independent samples.

Minor points:

1. In figure S3A, what is the substrate of the DNR mold?

Response: We thank the reviewer for the question. The substrate of the DNR mold in Figure S3A is glass. We have added the description in the caption of Figure S3A.

2. The abbreviation "Al" in Figure S3B requires clarification. Additionally, it is worth noting why materials such as PMMA and PTFE, presented in Figure S3A, are absent from Figure S3B. This omission raises questions about the selection of materials for presentation in the context of this comparison.

Response: We thank the reviewer for the reminder. We have used the full name of "Aluminum" instead of "Al" in Figure S3B. The absence of PMMA and PTFE in Figure S3A is because of the surface modification of the two substrates with organosilanes being highly challenging. As PMMA and PTFE are chemically inert, it is difficult to functionalize their surface even after the pretreatment by oxygen plasma. Therefore, we did not prepare DNR molds by PMMA and PTFE substrates.

We have added the explanation in the caption of Figure S3B.

3. Incorporating time stamps within Movie S1 and S2 would facilitate a more effective comparison between the two videos.

Response: We thank the reviewer for the suggestion. We have added the time stamps in the revised Movie S1 and S2.

4. In Figure S10B, there appears to be a generally higher intensity for the conventional hydrogel. Assuming this difference is for visualization purposes, it would be advisable to explicitly state this clarification within the figure caption.

Response: We thank the reviewer for the kind reminder. We have added relevant description in the caption of Figure S10B: "To make the contrast of XPS curves more distinctive, the overall intensity of DNR hydrogel was adjusted to being comparable with the conventional hydrogel."

5. To offer readers a comprehensive perspective, it could be useful to include a conventional hydrogel attachment to a porcine heart in addition to Movie S3. This addition would provide a visual contrast for readers to better understand the differences between the two scenarios.

Response: We thank the reviewer for the suggestion. We have added the adhesion circumstance of conventional hydrogel on a porcine heart in Movie S5.

6. Incorporating appropriate significance markings on the graph would enhance its readability. Currently, the error bars overlap between the 50 and 100 data points in Figure 2B without any indication, which might imply lack of statistical significance, which is a similar case in Figure 2E and Figure 2H. In Figure 3H, the main text claims there is a statistical significance, it is also not presented in the caption nor on the figure. In Figure 3H, the main text asserts the presence of statistical significance; however, this information is not indicated in the caption nor on the figure itself.

Response: We thank the reviewer for the helpful reminders and we are sorry for the confusion caused. We have added the significance markings in the revised Figure 2B, 2E and 2H. It shows the significant difference ($P = 0.004$) between the 50 and 100 data points in Figure 2B. In addition, we have added the statistical information (P -value) in the caption of Figure 3H, which shows the significant difference of the mean adhesion time of PAA.

7. It is advisable to include a mention of the mold substrate used for grafting hydrophobic silicone chains within the main text. Presently, readers must refer to caption of Figure S1b to acquire this information (glass mold), which could be more seamlessly integrated into the narrative.

Response: We thank the reviewer for the kind reminder. We have referred to the glass mold used for the grafting of hydrophobic silicone chains in paragraph 2 on page 4.

8. For clarity, the authors are encouraged to cite only those parts of the figures that are relevant to the statement being made. For instance, while discussing the use of the DNR strategy to maintain hydrogel's chemical composition and bulk properties, citing Figure 1B instead of Figure 1 could help to save the reader time.

Response: We thank the reviewer for the suggestion. We have properly amended our citations to the figures in the revised manuscript.

9. The authors should consider increasing the contrast difference in the colored figures to ensure that they remain distinguishable for readers with color blindness.

Response: We thank the reviewer for the thoughtful reminder. We have increased the contrast difference of the colored figures in the revised manuscript.

10. In Figure S9A, the inclusion of a title in Chinese could be addressed by either omitting it from the graph or providing a translation to English. This adjustment would ensure clarity and consistency in the presentation of the figure.

Response: We thank the reviewer for the reminder. We have provided the English translation of the Chinese title in the revised Figure S9A.

Reviewer #3:

This manuscript presents a novel gelation strategy called the ‘Interface-mediated dynamic network reconfiguration (DNR)’, which effectively transforms the surface hydrophilicity of hydrogels into a hydrophobic state while maintaining the chemical composition and overall characteristics of the hydrogels. The hydrogels prepared using DNR strategy exhibited substantially enhanced bioadhesion and improved transdermal delivery as compared with the hydrogels fabricated via the conventional method. The concept in this study is novel and fascinating, and the manuscript is concisely well written and provides many convincing data to support the authors’ claims. This reviewer recommends minor revision to address the following issues before publication.

Response: We highly appreciate the reviewer for the positive comments on our work and the constructive suggestions.

1. Figure S4 shows the surface hydrophobicity of the PAA-based DNR hydrogel after 24 hours in a humid environment. It would also be beneficial to provide the pictures of the hydrogel to show any changes in the shape or structure of the hydrogel in the humid environment.

Response: We thank the reviewer for the suggestion. We have added the pictures of the hydrogel in Figure S4B. It shows that the shape of hydrogel did not significantly change in the humid environment.

2. Figure S7 is showing fabrication of the Janus hydrogels with asymmetric wettability using the DNR strategy, which provides an interesting avenue of the study. Additional characterization of the Janus hydrogels to validate asymmetric wettability would be valuable information to the readers (e.g., WCA comparison in different side of the hydrogel).

Response: We thank the reviewer for the kind reminder. We have provided the WCA values in different side of the Janus hydrogel in Figure S7C.

3. Figure S13 demonstrates the bulk properties of the hydrogel. Regarding the swelling property of the hydrogels, the pictures of the hydrogels over time in swelling test would verify the maintenance of bulk property of the hydrogels in wet conditions.

Response: We thank the reviewer for the suggestion. We have added the pictures of the hydrogels over time in swelling test in the revised Figure S14E.

4. In Figure 2B, the authors mentioned that the amount of crosslinker did not significantly affect the wettability of the DNR mold. However, the graph of Figure 2B seems to show some notable differences in the WCA of molds. The difference is not statistically significant?

Response: We thank the reviewer for this important question and we are sorry for the

improper description of the wettability of the DNR mold. The difference is indeed significant, and we have added the statistical significance markings to the WCA of molds in the revised Figure S16.

We have corrected our description in paragraph 3 on page 8.

5. Figure 2 provides important data since it nicely demonstrates that the structural dynamics of silicone chains grafted on the mold surface are crucial to reconfigure hydrogel surface networks. This figure indicates that silicon crosslinking degree, chain length, and chain density show different effects on gelation process of DNR hydrogels and surface properties of the resultant DNR hydrogels. Thus, this reviewer recommends the authors to elaborate further discussion on feasible underlying chemical mechanisms of these factors impacting on the surface properties of DNR hydrogels.

Response: We thank the reviewer for this constructive suggestion. To elucidate the underlying chemical mechanism of the effect of silicone chain crosslinking degree on surface properties of DNR hydrogels, we performed XPS analysis on the hydrogels prepared by DNR molds with various crosslinking degree of silicone chain (Figure S17). The carbon content in the hydrogel surface declined from 74 atom% to 69 atom% when the MTMS content increased from 0 wt% to 50 wt% (Figure S17A and S17B), which is very close to the value of the conventional hydrogel (68 atom%). Moreover, the gradual decrease of hydrophobic segments in the hydrogel surface demonstrates the weakened surface network reconfiguration of DNR hydrogel (Figure S17C and S17D). These results corroborate the inhibition of the DNR effect with increased silicone chain crosslinking degree on the DNR mold, thus impacting the surface properties of DNR hydrogels. In addition, we reason that a shorter length and higher density of silicone chains on DNR molds could also impact the surface properties of DNR hydrogels by compromising the DNR effect in the hydrogel surface and reducing the contents of carbon and hydrophobic segments.

We have added the discussions in paragraph 3 on page 8 and paragraph 2 on page 9.

Fig. S17. (A) XPS spectra of DNR hydrogels prepared by the DNR molds with different crosslinking degrees of silicone chain. (B) Element contents of carbon and oxygen on the DNR hydrogel surface. The values were determined by integration of the C 1s and O 1s peak areas in A. (C) High-resolution XPS spectra of DNR hydrogels prepared by the DNR molds with different crosslinking degrees of silicone chain. (D) Statistics of the element content distribution of polymer segments in the DNR hydrogels.

6. Figure 3 is showing the molecular dynamics simulations to analyze the structural and dynamic details of the PAA hydrogels. I just wonder whether other types of polymers mentioned in Figure 1B (PMAA, PAAM) exhibit similar dynamic behaviors when fabricated through DNR protocol.

Response: We thank the reviewer for the question. To address the reviewer's question, we have conducted multiple 1- μ s MD simulations to investigate the dynamics of PMAA and PAAM polymers (25 units). Specifically, we first placed a single PMAA or PAAM chain on DNR molds with either 0%, 50% or 100% silicone chains restrained, consistent with the settings of our PAA simulations reported in Fig. 3E and Fig. 3F. As shown below, these simulations indicate that similar to PAA, PMAA exhibits a significant reduction in dynamics on the restrained DNR molds, and the extent of such reduction is even greater than that observed for PAA, which may be attributed to the greater hydrophobicity of PMAA. For PAAM, however, inadequate adhesion of a single PAAM chain onto the silicone chain-grafted mold prevented us from collecting sufficient statistics to measure the diffusion coefficient of single PAAM along the mold surface. This is likely attributed to the greater hydrophilicity of the amide group of PAAM than the carboxyl group of PAA. In light of this result, we next simulated 60 PAAM chains with an initial random placement in water (see figure below), similar to the setting for multi-chain PAA simulation as reported in Fig. 3A. Subsequent 1- μ s

simulation reveals that PAAm indeed adheres to DNR mold surface in this multi-chain setting. Therefore, quantifying the impact of DNR molds on the dynamics of PAAm likely requires a larger length scale as well as a longer timescale than that of PAA and PMAA simulations, which we plan to further investigate in a dedicated simulation study.

7. Figure 4 nicely presents application of DNR hydrogel for wet bioadhesion and transdermal delivery. If the authors consider biomedical applications, biocompatibility of the DNR hydrogel, such as cytotoxicity and inflammation, should be investigated. Also, drug release profiles from DNR hydrogel may need to be examined under physiologically relevant conditions.

Response: We thank the reviewer for the helpful suggestions. We examined the biocompatibility of DNR hydrogel by analysis of its cytotoxicity and inflammatory response (Figure S34). The results of MTT assay and live/dead staining showed that the majority of the L929 cells cultured with the DNR hydrogel extracts remained viable as also found in the control group (Figure S34A and S34B). Next, DNR hydrogel was subcutaneously implanted in mice for 14 days to evaluate its inflammatory reaction. Hematoxylin-eosin (H&E) staining showed that the amount of inflammatory cells in DNR hydrogel group was similar to that in control group (Figure S34C), suggesting the good in vivo biocompatibility of DNR hydrogel. Moreover, the blood biochemical indicators of mice receiving DNR hydrogel implantation, including alanine aminotransferase (ALT), aspartate aminotransferase (AST), blood urea nitrogen (BUN), and creatinine (CREA), showed no significant difference compared with that of the control group (Figure S34D), indicating the low organ toxicity of DNR hydrogel. These results imply the good biocompatibility of DNR hydrogel. We have added the experimental procedures to “In vitro cytocompatibility” and “In vivo biocompatibility” sections on page 25 and the discussion in paragraph 2 on page 18.

Fig. S34. Evaluation of the biocompatibility of DNR hydrogel. **(A)** Cytotoxicity of DNR hydrogel by MTT method. The L929 cells were cultured in DMEM with different concentration of hydrogel extracts for different periods. Control group: 0 mg/mL; DNR hydrogel group: 1–5 mg/mL. **(B)** Representative images of L929 cells with live/dead staining after incubation with different concentration of hydrogel extracts for 5 days (scale bar: 100 μ m; control group: 0 mg/mL; DNR hydrogel group: 1–5 mg/mL). **(C)** Hematoxylin-eosin (H&E) staining images of skin tissue after subcutaneous embedding of DNR hydrogels in Balb/C mouse skin incision model for 14 days ($N = 4$ biologically independent samples in each group; scale bar: 300 μ m). **(D)** Blood biochemistry analysis of liver and kidney function markers (ALT, AST, BUN, and CREA). Liver function assessment: ALT (alanine aminotransferase), AST (aspartate aminotransferase), BUN (blood urea nitrogen). Kidney function assessment: CREA (creatinine). Values in **A** and **D** are presented as the mean \pm SD, $N = 4$ –6 independent samples.

We studied the drug release profile of the DNR hydrogel under physiologically relevant conditions by using sodium fluorescein as the model drug (Figure S33). The DNR hydrogel demonstrates a burst release with 1 hour and a subsequent steady release up to 20 hours with the cumulative release rate of \sim 18%. We have added the experimental procedures to “In vitro drug release” section on page 24 and the discussion in paragraph 1 on page 18.

Fig. S33. (A) The standard curve of sodium fluorescein, which was determined by a UV-vis spectrophotometer at 474 nm. (B) The cumulative release of sodium fluorescein from the DNR hydrogel in PBS at 37 °C. Values in B are shown as the mean \pm SD; $N = 3$ independent samples.

8. Please provide discussion to compare advantages and disadvantages between DNR hydrogels and well-known marine-inspired adhesive hydrogels functionalized with phenolic moieties (e.g., catechol, gallol).

Response: We thank the reviewer for the valuable suggestion. Compared to the well-known marine-inspired adhesive hydrogels functionalized with phenolic moieties⁷⁻¹⁰, our DNR strategy may demonstrate a series of advantages including free of intricate chemical synthesis, ease of implementation, and wide applicability. Nevertheless, there are also limitations for the DNR hydrogels, such as the matrix polymer-dependent adhesion performance, susceptibility to the surface wettability transition, and possible batch-to-batch variations because of the repeated use of DNR molds.

We have added the discussion in paragraph 1 on page 16.

References

1. Ligon, S.C., Husár, B., Wutzel, H., Holman, R. & Liska, R. Strategies to Reduce Oxygen Inhibition in Photoinduced Polymerization. *Chem. Rev.* **114**, 557-589 (2014).
2. Zhao, Z., Zhang, K., Liu, Y., Zhou, J. & Liu, M. Highly Stretchable, Shape Memory Organohydrogels Using Phase-Transition Microinclusions. *Adv. Mater.* **29**, 1701695 (2017).
3. Zhuo, S. et al. Complex multiphase organohydrogels with programmable mechanics toward adaptive soft-matter machines. *Sci. Adv.* **6**, eaax1464 (2020).
4. Meier, Y.A., Zhang, K., Spencer, N.D. & Simic, R. Linking Friction and Surface Properties of Hydrogels Molded Against Materials of Different Surface Energies. *Langmuir* **35**, 15805-15812 (2019).
5. Gombert, Y. et al. Structuring Hydrogel Surfaces for Tribology. *Adv. Mater. Interfaces* **6**, 1901320 (2019).
6. Simič, R., Mandal, J., Zhang, K. & Spencer, N.D. Oxygen inhibition of free-radical polymerization is the dominant mechanism behind the “mold effect” on hydrogels. *Soft Matter* **17**, 6394-6403 (2021).
7. Maier, G.P., Rapp, M.V., Waite, J.H., Israelachvili, J.N. & Butler, A. Adaptive synergy between catechol and lysine promotes wet adhesion by surface salt displacement. *Science* **349**, 628-632 (2015).
8. Zhang, W. et al. Catechol-functionalized hydrogels: biomimetic design, adhesion mechanism, and biomedical applications. *Chem. Soc. Rev.* **49**, 433-464 (2020).
9. Narayanan, A., Dhinojwala, A. & Joy, A. Design principles for creating synthetic underwater adhesives. *Chem. Soc. Rev.* **50**, 13321-13345 (2021).
10. Gan, D. et al. Plant-inspired adhesive and tough hydrogel based on Ag-Lignin nanoparticles-triggered dynamic redox catechol chemistry. *Nat. Commun.* **10**, 1487 (2019).

REVIEWERS' COMMENTS

Reviewer #1 (Remarks to the Author):

The authors have made proper modification, and I believe that this manuscript can be accepted for publication in Nature Communications in the current form.

Reviewer #2 (Remarks to the Author):

The authors have adequately responded to my concerns.

Reviewer #3 (Remarks to the Author):

The authors have properly addressed all issues raised by this reviewer and revised the manuscript accordingly. Indeed, the manuscript has been substantially improved through revision. I appreciate the authors' hard work for revision. Please check minor comments below and then I would recommend acceptance of the revised version of the manuscript for publication.

1. In Figure S34C, there is no sign of DNR hydrogel in the H&E image. Did the implanted DNR hydrogel completely degrade in vivo within 14 days? Please provide speculation on the degradability of the DNR hydrogel.
2. In response to my comment 8, please further elaborate the principles behind the advantages of DNR hydrogel compared with marine-inspired phenolic hydrogels.

Response to the Comments from the Reviewers:

We gratefully acknowledge the reviewers' encouraging comments and feedback on our manuscript. Here we provide a point-by-point response to each of the reviewers. The corresponding changes to the revised manuscript have been highlighted in yellow.

Reviewer #1:

The authors have made proper modification, and I believe that this manuscript can be accepted for publication in Nature Communications in the current form.

Response: We sincerely appreciate the reviewer's recognition of our manuscript.

Reviewer #2:

The authors have adequately responded to my concerns.

Response: We heartily thank the reviewer again for the positive comments and valuable suggestions to improve the quality of our manuscript.

Reviewer #3:

The authors have properly addressed all issues raised by this reviewer and revised the manuscript accordingly. Indeed, the manuscript has been substantially improved through revision. I appreciate the authors' hard work for revision. Please check minor comments below and then I would recommend acceptance of the revised version of the manuscript for publication.

Response: We gratefully thank the reviewer again for the positive comments and feedback on our manuscript.

1. In Figure S34C, there is no sign of DNR hydrogel in the H&E image. Did the implanted DNR hydrogel completely degrade in vivo within 14 days? Please provide speculation on the degradability of the DNR hydrogel.

Response: We thank the reviewer for the question and we are sorry for the confusion caused. The DNR hydrogel implants have been removed from the tissue before the histological processing and H&E staining to better show the imaged tissues.

We have added the relevant description in the "In vivo biocompatibility" section on page 20.

2. In response to my comment 8, please further elaborate the principles behind the advantages of DNR hydrogel compared with marine-inspired phenolic hydrogels.

Response: We thank the reviewer for the constructive suggestion. The adhesion mechanism of our DNR hydrogel is based on the physical repelling of interfacial water

by the hydrophobic hydrogel surface network and therefore enhancing surface contact between the hydrogel network and substrate, while the adhesion of phenolic hydrogels is based on the chemical bonding with the substrates. Both strategies of fabricating the bioadhesive hydrogels have its own advantages and limitations. Compared with the chemical bonding-based strategy, our DNR strategy may demonstrate a series of advantages including free of intricate chemical synthesis, ease of implementation, and wide applicability. Nevertheless, there are also limitations for the DNR hydrogels, such as the matrix polymer-dependent adhesion performance, susceptibility to the surface wettability transition, and possible batch-to-batch variations because of the repeated use of DNR molds.